# WEIGHTED ENSEMBLE SELF-SUPERVISED LEARNING

**Yangjun Ruan**[*†]    **Saurabh Singh**    **Warren Morningstar**    **Alexander A. Alemi**

**Sergey Ioffe**    **Ian Fischer**[†]    **Joshua V. Dillon**[†]

Google Research

## ABSTRACT

Ensembling has proven to be a powerful technique for boosting model performance, uncertainty estimation, and robustness in supervised learning. Advances in self-supervised learning (SSL) enable leveraging large unlabeled corpora for state-of-the-art few-shot and supervised learning performance. In this paper, we explore how ensemble methods can improve recent SSL techniques by developing a framework that permits data-dependent *weighted* cross-entropy losses. We refrain from ensembling the representation backbone; this choice yields an efficient ensemble method that incurs a small training cost and requires no architectural changes or computational overhead to downstream evaluation. The effectiveness of our method is demonstrated with two state-of-the-art SSL methods, DINO (Caron et al., 2021) and MSN (Assran et al., 2022). Our method outperforms both in multiple evaluation metrics on ImageNet-1K, particularly in the few-shot setting. We explore several weighting schemes and find that those which increase the diversity of ensemble heads lead to better downstream evaluation results. Thorough experiments yield improved prior art baselines which our method still surpasses; e.g., our overall improvement with MSN ViT-B/16 is 3.9 p.p. for 1-shot learning.

## 1 INTRODUCTION

The promise of self-supervised learning (SSL) is to extract information from unlabeled data and leverage this information in downstream tasks (He et al., 2020; Caron et al., 2021); e.g., semi-supervised learning (Chen et al., 2020a;b), robust learning (Radford et al., 2021; Ruan et al., 2022; Lee et al., 2021), few-shot learning (Assran et al., 2022), and supervised learning (Tomasev et al., 2022). These successes have encouraged increasingly advanced SSL techniques

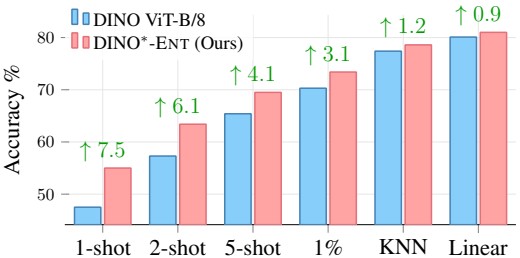

Figure 1: Our improvements to DINO, including baseline improvements and ensembling.

(e.g., Grill et al., 2020; Zbontar et al., 2021; He et al., 2022). Perhaps surprisingly however, a simple and otherwise common idea has received limited consideration: ensembling.

Ensembling combines predictions from multiple trained models and has proven effective at improving model accuracy (Hansen & Salamon, 1990; Perrone & Cooper, 1992) and capturing predictive uncertainty in supervised learning (Lakshminarayanan et al., 2017; Ovadia et al., 2019). Ensembling in the SSL regime is nuanced, however; since the goal is to learn useful representations from unlabeled data, it is less obvious *where* and *how* to ensemble. We explore these questions in this work.

We develop an efficient ensemble method tailored for SSL that replicates the non-representation parts (e.g., projection heads) of the SSL model. In contrast with traditional "post-training" ensembling, our ensembles are only used during training to facilitate the learning of a *single* representation encoder, which yields no extra cost in downstream evaluation. We further present a family of weighted cross-entropy losses to effectively train the ensembles. The key component of our losses is the introduction of *data-dependant importance weights* for ensemble members. We empirically compare different choices from our framework and find that the choice of weighting schemes critically impacts ensemble diversity, and that greater ensemble diversity correlates with improved downstream performance. Our method is potentially applicable to many SSL methods; we focus on DINO (Caron et al., 2021) and MSN (Assran et al., 2022) to demonstrate its effectiveness. Fig. 1 shows DINO improvements from using our ensembling and weighted cross-entropy loss.

---

[*]University of Toronto & Vector Institute. Work done as a student researcher at Google.

[†]Correspondence to yjruan@cs.toronto.edu, {iansf, jvdillon}@google.com.

In summary, our core contributions are to:

- Develop a downstream-efficient ensemble method suitable for many SSL techniques (Sec. 3.1).
- Characterize an ensemble loss family of *weighted* cross-entropy objectives (Sec. 3.2).
- Conduct extensive ablation studies that improve the prior art baselines by up to 6.3 p.p. (Sec. 5.1).
- Further improve those baselines with ensembling (e.g., up to 5.5 p.p. gain for 1-shot) (Table 2).

## 2 BACKGROUND

In this section, we frame SSL methods from the perspective of maximum likelihood estimation (MLE) and use this as the notational basis to describe the state-of-the-art clustering-based SSL methods as well as derive their ensembled variants in Sec. 3.

**From Maximum Likelihood to SSL**  Denote unnormalized KL divergence (Dikmen et al., 2014) between non-negative integrable functions $p, q$ by $\mathsf{K}[p(X), q(X)] = \mathsf{H}^\times[p(X), q(X)] - \mathsf{H}[p(X)]$, where $\mathsf{H}^\times[p(X), q(X)] = -\int_\mathcal{X} p(x) \log q(x) \mathrm{d}x + \int_\mathcal{X} q(x) \mathrm{d}x - 1$ is the unnormalized cross-entropy (with $0 \log 0 = 0$) and $\mathsf{H}[p(X)] = \mathsf{H}^\times[p(X), p(X)]$. These quantities simplify to their usual definitions when $p, q$ are normalized, but critically they enable flexible weighting of distributions for the derivation of our weighted ensemble losses in Sec. 3.2.

Let $\nu(X, Y) = \nu(X)\nu(Y|X)$ be nature's distribution of input/target pairs over the space $\mathcal{X} \times \mathcal{Y}$ and $s(Y|\theta, X)$ be a predictive model of target given the input parameterized by $\theta \in \mathcal{T}$. Supervised maximum likelihood seeks the minimum expected conditional population risk with respect to $\theta$,

$$\mathsf{E}_{\nu(X)} \mathsf{K}[\nu(Y|X), s(Y|\theta, X)] = \mathsf{E}_{\nu(X)} \mathsf{H}^\times[\nu(Y|X), s(Y|\theta, X)] - \mathsf{E}_{\nu(X)} \mathsf{H}[\nu(Y|X)]. \quad (1)$$

Henceforth omit $\mathsf{E}_{\nu(X)} \mathsf{H}[\nu(Y|X)]$ since it is constant in $\theta$. Since $\nu(X, Y)$ is unknown, a finite sample approximation is often employed. Denote a size-$n$ i.i.d. training set by $\mathcal{D}_n = \{x_i\}_{i\in[n]} \sim \nu^{\otimes n}$ and empirical distribution by $\hat{\nu}(X, Y) = \frac{1}{n} \sum_{x \in \mathcal{D}_n, y \sim \nu(Y|x)} \delta(X - x)\delta(Y - y)$ where $\delta : \mathbb{R} \to \{0, 1\}$ is 1 when $x = 0$ and 0 otherwise. The sample risk is thus $-\frac{1}{n} \sum_{x \in \mathcal{D}_n} \mathsf{H}^\times[\hat{\nu}(Y|x), s(Y|\theta, x)]$.

In SSL, we interpret $\nu(Y|x)$ as being the oracle teacher under a presumption of how the representations will be evaluated on a downstream task. This assumption is similar to that made in Arora et al. (2019); Nozawa et al. (2020). We also assume $\hat{\nu}(Y|X)$ is inaccessible and/or unreliable. Under this view, some SSL techniques substitute $\hat{\nu}(Y|x)$ for a weakly learned target or "teacher", $t(Y|x)$. We don't generally expect $t(Y|x)$ to recover $\nu(Y|x)$; we only assume that an optimal teacher exists and it is $\nu(Y|x)$. With the teacher providing the targets, the loss becomes $-\frac{1}{n} \sum_{x \in \mathcal{D}_n} \mathsf{H}^\times[t(Y|x), s(Y|\theta, x)]$.

**Teacher and student in clustering SSL methods**  Clustering SSL methods such as SWaV (Caron et al., 2020), DINO (Caron et al., 2021), and MSN (Assran et al., 2022) employ a student model characterized by proximity between learned codebook entries and a data-dependent code,

$$s(Y|\theta, x) = \mathrm{softmax}\left(\left\{\frac{1}{\tau} \frac{(h_\psi \circ r_\omega)(x) \cdot \mu_y}{\|(h_\psi \circ r_\omega)(x)\|_2 \|\mu_y\|_2} : y \in [c]\right\}\right) \quad (2)$$

$$\theta = \{\omega, \psi, \{\mu_y\}_{y\in[c]}\} \in \mathcal{T}, \quad (3)$$

where the encoder $r_\omega : \mathcal{X} \to \mathcal{Z}$ produces the representations used for downstream tasks, and the projection head $h_\psi : \mathcal{Z} \to \mathbb{R}^d$ and codebook entries $\{\mu_y\}_{y\in\mathcal{Y}} \in \mathbb{R}^d$ characterize the SSL loss. Eq. (2) can be viewed as "soft clustering", where the input is assigned to those centroids that are closer to the projection head's output. The projection head and codebook are used during training but thrown away for evaluation, which is empirically found vital for downstream tasks (Chen et al., 2020a;b). Hyperparameters $\tau \in \mathbb{R}_{>0}, c \in \mathbb{Z}_{>0}$ represent temperature and codebook size. The teacher is defined as $t(Y|x) = s(Y | \mathrm{stopgrad}(g(\theta)), x)$ where $g : \mathcal{T} \to \mathcal{T}$. Commonly $g(\theta)$ is an exponential moving average of gradient descent iterates and the teacher uses a lower temperature than the student.

To capture desirable invariances and prevent degeneracy, data augmentation and regularization (e.g., Sinkhorn-Knopp normalization (Caron et al., 2020), mean entropy maximization (Assran et al., 2022)) are essential. As these are not directly relevant to our method, we omit them for brevity.

## 3 METHOD

Ensembling is a technique that combines models to boost performance, and has been especially successful in supervised learning. We are interested in ensembling methods that carry over this success to SSL approaches. However, SSL has key differences, such as throw-away "projection heads", from supervised learning that result in a multitude of possibilities for how to ensemble. With this in mind, we propose first *where* to ensemble, and then *how* to ensemble. Those proposals result in an efficient "peri-training" ensembling technique specifically tailored for SSL and a family of *weighted* ensemble objectives; we subsequently suggest different ways to select the weights.

### 3.1 WHERE TO ENSEMBLE?

Denote the teacher/student ensembles by $\{t_i(Y|x)\}_{i\in[m]}$ and $\{s(Y|\theta_j,x)\}_{j\in[m]}$ and define each as in Sec. 2; parameters $\theta = \{\theta_j\}_{j\in[m]} \in \mathcal{T}^m$ are independently initialized, all students use one temperature and all teachers another. We asymmetrically denote $t_i(Y|x)$ and $s(Y|\theta_j,x)$ to emphasize that teachers' gradients are zero and that the students are distinct solely by way of $\theta_i \neq \theta_j$. Studying heterogeneous architectures and/or different teacher parameterizations is left for future work.

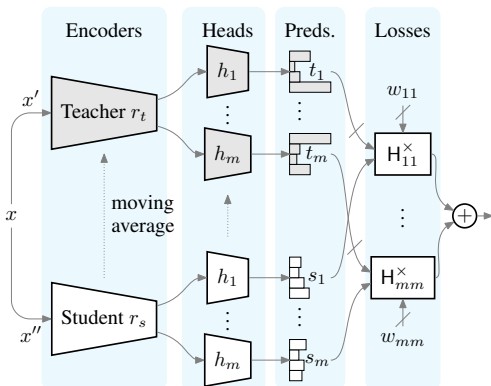

Recall that $\theta_j$ parameterizes the encoder, projection head, and codebook parameters: $\theta_j = (\omega_j, \psi_j, \{\mu_{jy}\}_{y\in\mathcal{Y}})$. We further restrict $\mathcal{T}^m$ such that $\omega_i = \omega_j$, i.e., we limit our consideration to ensembles of projection heads $h_{\psi_j}$ and/or codebooks $\mu_j$ but not encoders $r_{\omega_j}$. This choice makes our ensemble method inherently different from traditional supervised ensembling or encoder $r_\omega$ ensembling: the ensembled parts are not used for evaluation but

Figure 2: **Overview of $(h_\psi, \mu)$-ensemble.** Two augmented inputs are encoded by the teacher/student into representations, and then processed by an ensemble of heads. The loss for each head is weighted and summed into the final loss. Strike-through edges indicate stop-gradients. See Appx. A for pseudocode.

for improving the learning of non-ensembled representation encoder during training, thus it requires no change of downstream evaluation or computational cost. Ensembling of $r_\omega$ is left for future work.

### 3.2 HOW TO ENSEMBLE?

We would like to extend the loss to support an ensemble of teacher/student pairs while respecting the MLE intuition of the loss as in Sec. 2. Additionally, we want to facilitate data-dependent importance weights, thus enabling preferential treatment of some teacher/student pairs. We therefore propose a weighted average (unnormalized) cross-entropy loss,

$$\mathcal{L}_n(\theta) = \frac{1}{n} \sum_{x\in\mathcal{D}_n} \sum_{i,j\in[m]} \mathsf{H}^\times[w_{ijY} \odot t_i(Y|x), s(Y|\theta_j,x)] \tag{4}$$

$$\text{where} \quad w_{ijy} = \text{softmax}\left(\left\{\tfrac{1}{\gamma}f_{ijy}(\text{stopgrad}(\theta),x) : i,j\in[m]\right\}\right). \tag{5}$$

The notation $w_{ijY} \odot t_i(Y|x)$ denotes a Hadamard product; i.e., the product of event-specific weights and probabilities for each $y \in \mathcal{Y}$. The hyperparameter $\gamma$ is the temperature. The function $f_{ijy}$ is defined for brevity and discussed in the following section.

This objective admits generality and flexibility for introducing various weighting schemes, as it supports potential interactions between all teacher/student pairs and allows the weights to be both model- and data-dependent. Up to a constant independent of $\theta$, it is an importance weighted average of (unnormalized) KL divergences between each teacher and each student; i.e., a mixture of MLE-like objectives. We stop the gradient of $w_{ijy}$ to $\theta$ in order to keep the overall gradient a weighted average of students' log-likelihood gradients, similar to Eq. (1). We also normalize the weights such that each data point equally contributes to the loss.

### 3.3 HOW TO WEIGHT?

In this section, we present several instantiations of our losses with different weighting schemes. We empirically show in Sec. 5 that the particular choice of weighting scheme is critical for the representation performance and the induced diversity of $(h_\psi, \mu)$-ensembles. For simplicity we assume $\gamma = 1$ in this section. We indicate with $\iff$ that a loss has the same $\arg\min$ as Eq. (4). For additional analysis and discussion, see Appx. D.

**Uniform weighting (UNIF)** The simplest strategy is to treat different teacher/student pairs independently and average each with uniform weighting; i.e.,

$$f_{ijy} = \log \delta(i - j) \iff \mathcal{L}_n^{\text{UNIF}}(\theta) = \frac{1}{n} \sum_{x \in \mathcal{D}_n} \frac{1}{m} \sum_{i \in [m]} \mathsf{H}^\times [t_i(Y|x), s(Y|\theta_i, x)] \tag{6}$$

This strategy introduces *uniform* weights $w_i = \frac{1}{m}$ over ensemble elements. The role of $\log \delta(i - j)$ (here and elsewhere) is to sub-select corresponding teacher/student pairs rather than all $m^2$ pairs.

**Probability weighting (PROB)** An alternative to using the average cross-entropy loss (UNIF) is to compute the cross-entropy loss of the average predictions whose gradient is weighted by $w_{ijy}$ (see Appx. D.1). At $\gamma = 1$, those gradient weights simplify into an average over the student probabilities:

$$f_{ijy} = \log s(y|\theta_j, x) \iff \mathcal{L}_n^{\text{PROB}}(\theta) = \frac{1}{n} \sum_{x \in \mathcal{D}_n} \mathsf{H}^\times \left[ \frac{1}{m} \sum_{i \in [m]} t_i(Y|x), \frac{1}{m} \sum_{j \in [m]} s(Y|\theta_j, x) \right] \tag{7}$$

Averaging the predictive distributions introduces correspondence between codes from different heads; thus different heads are no longer independent but instead *cooperate* to match the student to the teachers. The loss favors student heads with more confident predictions (i.e., larger $s(y|\theta_j, x)$). Further motivation for averaging predictions comes from multi-sample losses studied in Morningstar et al. (2022). Note that the joint convexity of (unnormalized) KL divergence implies that this loss is upper bounded by the UNIF loss up to some constant in $\theta$ (see Appx. D).

Although the PROB strategy favors confident student predictions, the weights change as a function of $y \in \mathcal{Y}$. This may be in conflict with our intuition that SSL is like maximum likelihood (Sec. 2), since under that view, the teacher is responsible for weighting outcomes.

**Entropy weighting (ENT)** Another way to favor heads with more confident predictions is to directly weight by their predictive entropies; i.e.,

$$f_{ijy} = - \mathsf{H}[t_i(Y|x)] + \log \delta(i - j) \iff \tag{8}$$

$$\mathcal{L}_n^{\text{ENT}}(\theta) = \frac{1}{n} \sum_{x \in \mathcal{D}_n} \sum_{i \in [m]} \operatorname{softmax}_i(\{-\tfrac{1}{\gamma} \mathsf{H}[t_{i'}(Y|x)] : i' \in [m]\}) \, \mathsf{H}^\times [t_i(Y|x), s(Y|\theta_i, x)] \tag{9}$$

where the weight $w_i = \operatorname{softmax}_i(\{-\frac{1}{\gamma} \mathsf{H}[t_{i'}(Y|x)] : i' \in [m]\})$ is inversely correlated with the entropy of teacher predictions. In other words, the head whose teacher has a lower entropy (i.e., higher confidence about its prediction) is given a larger importance weight for learning the representation. Like PROB, this strategy encourages "data specialists" by emphasizing strongly opinionated teacher heads for different inputs. Like UNIF, different heads are treated more independent (than PROB), since interaction between different heads is introduced only through the weight computation. By preferring low-entropy teachers we also favor low variance teachers; this aligns with the intuition that using a lower-variance teacher benefits representation quality (Wang et al., 2022).

**Countless other weighting schemes** It is impossible to fully explore the space of weightings; the following might also be interesting to study in detail but were omitted due to resource constraints.

$$
\begin{array}{lll}
f_{ijy} = 0 & \text{(Favors all pairs of teachers/students equally)} & (10) \\
f_{ijy} = \log t_i(y|x) & \text{(Favors opinionated teachers)} & (11) \\
f_{ijy} = - \mathsf{H}[s(Y|\theta_j, x)] & \text{(Favors low-entropy students)} & (12) \\
f_{ijy} = \mathsf{K}[t_i(Y|x), s(Y|\theta_j, x)] & \text{(Favors disagreeing teacher/student pairs)} & (13)
\end{array}
$$

$$f_{ijy} = -\tfrac{1}{2}\log(\mathrm{Var}_{t_i(Y|x)}[Y] + \epsilon) \qquad \text{(Favors low variance teachers; e.g., } \epsilon = \tfrac{1}{12}) \qquad (14)$$

Note that "aligned" versions of all schemes are possible by using $f_{ijy} + \log\delta(i-j)$. We did early experiments exploring Eqs. (11) and (12), but the results were inferior and are largely omitted below.

## 4 RELATED WORK

**Self-supervised learning**   Recent work on self-supervised learning (SSL) focuses on discriminative or generative approaches. Most discriminative approaches seek to learn augmentation-invariant representations by enforcing the similarity between augmented pairs of the same image while utilizing different techniques to avoid collapse. Contrastive methods (Chen et al., 2020a; He et al., 2020; Wu et al., 2018; Hjelm et al., 2018; Bachman et al., 2019; Tian et al., 2020) use a large number of negative samples with a noise-contrastive objective (Gutmann & Hyvärinen, 2010; Oord et al., 2018). A large body of followup work eliminates the necessity of explicit negative samples with various techniques, including clustering assignment constraints (Caron et al., 2018; 2020; 2021; Asano et al., 2019), bootstrapping (Grill et al., 2020) or self-distillation (Caron et al., 2021) inspired by mean teacher (Tarvainen & Valpola, 2017), asymmetric architecture design (Grill et al., 2020; Chen & He, 2021), or redundancy reduction (Zbontar et al., 2021; Bardes et al., 2021). Recent generative approaches that use masked image modeling as the pretraining task (Dosovitskiy et al., 2020; Bao et al., 2021; He et al., 2022; Zhou et al., 2022; Xie et al., 2022) have achieved competitive finetuning performance. Our method may be applicable to all of the above methods that have some sort of "projection head", such as most of the discriminative approaches.

**Ensemble methods**   Ensembling has been extensively studied for improving model performance (Hansen & Salamon, 1990; Perrone & Cooper, 1992; Dietterich, 2000) and uncertainty estimation (Lakshminarayanan et al., 2017; Ovadia et al., 2019) in supervised learning and semi-supervised learning (Laine & Aila, 2016). A major research direction is to train efficient ensembles with partial parameter sharing (Lee et al., 2015; Wen et al., 2020; Dusenberry et al., 2020; Havasi et al., 2020) or intermediate checkpointing (Huang et al., 2017; Garipov et al., 2018). Our method also shares the encoder parameters across ensembles, which is closely related to multi-headed networks (Lee et al., 2015; Tran et al., 2020). Ensemble methods for SSL are less explored. Some recent work studies ensembles of supervised models adapted from pretrained SSL models. Gontijo-Lopes et al. (2022) conduct an empirical study of ensembles adapted from different SSL models and find that higher divergence in SSL methods leads to less correlated errors and better performance. Wortsman et al. (2022) ensemble multiple finetuned models adapted from the same SSL model by averaging their weights, which boosts the performance without any inference cost. Our method differs from them in that it (1) applies to the SSL training stage to directly improve representation quality, rather than aggregates multiple models in the post-training/finetuning stage; (2) introduces little training cost and no evaluation cost; and (3) is complementary to these post-training/finetuning ensembling methods.

## 5 EXPERIMENTS

We carefully study the impact of $(h_\psi, \mu)$-ensembles and our selected weighted ensemble losses (UNIF, PROB, and ENT) on smaller DINO models in Sec. 5.1. Using what we learned in those experiments, in Sec. 5.2 we present new state-of-the-art results on ImageNet-1K on various metrics for multiple model sizes by ensembling both DINO- and MSN-based models. Finally, we explore ensemble evaluations in the transfer learning setting in Sec. 5.3. Additional experimental details and results are in Appx. B and Appx. C, respectively.

**Experimental setup**   We assessed the effectiveness of our method with two SSL methods: DINO (Caron et al., 2021) and MSN (Assran et al., 2022). In order to ensure that we are comparing against strong baselines, we consider three different classes of baselines: **(1)** evaluation numbers reported in the original works (Caron et al. (2021), Assran et al. (2022), and Zhou et al. (2022) for an additional baseline iBOT); **(2)** evaluation of our implementation using the hyperparameters reported in the original works (DINO only, for space reasons) to validate our implementation; and **(3)** evaluation of our implementation using the best hyperparameters that we found by tuning the baselines (DINO and MSN) for fair comparisons. In almost all models and evaluations, our retuned baselines give non-trivial performance improvements on top of previously reported numbers. These type **(3)** baselines

Table 1: **Comparison of different ensemble strategies.** ENT and PROB significantly improve over the non-ensembled baseline, while UNIF leads to no gains. Ensembling both the projection head and the codebook works the best. All models are DINO* ViT-S/16 trained for 300 epochs. Averages and standard deviations are over 3 initialization seeds. The linear evaluation results on ImageNet-1K with different amounts of labeled data are reported here (see Table 11 in Appx. C.3 for all metrics).

| How | Where | | # of Labels Per Class | | | |
|---|---|---|---|---|---|---|
| | Proj. $h_\psi$ | Code. $\mu$ | 1 | 5 | ~13 (1%) | Full |
| Base | | | $40.6 \pm 0.2$ | $57.9 \pm 0.3$ | $63.4 \pm 0.2$ | $74.4 \pm 0.1$ |
| UNIF | ✓ | ✓ | $40.4 \pm 0.4$ | $57.6 \pm 0.3$ | $63.3 \pm 0.3$ | $74.5 \pm 0.2$ |
| PROB | ✓ | | $39.8 \pm 0.5 \downarrow 0.9$ | $57.4 \pm 0.4 \downarrow 0.5$ | $63.0 \pm 0.4 \downarrow 0.4$ | $74.8 \pm 0.1 \uparrow 0.4$ |
| PROB | ✓ | ✓ | $41.9 \pm 0.3 \uparrow 1.3$ | $59.6 \pm 0.4 \uparrow 1.7$ | $65.1 \pm 0.3 \uparrow 1.7$ | $\mathbf{75.4 \pm 0.1} \uparrow 1.0$ |
| ENT-ST | ✓ | ✓ | $40.0 \pm 0.5 \downarrow 0.6$ | $57.3 \pm 0.5 \downarrow 0.6$ | $62.7 \pm 0.5 \downarrow 0.7$ | $74.0 \pm 0.4 \downarrow 0.4$ |
| ENT | | ✓ | $40.8 \pm 0.4$ | $58.0 \pm 0.4$ | $63.5 \pm 0.4$ | $74.5 \pm 0.3$ |
| ENT | ✓ | | $43.0 \pm 0.6 \uparrow 2.4$ | $59.7 \pm 0.7 \uparrow 1.8$ | $64.8 \pm 0.5 \uparrow 1.4$ | $75.1 \pm 0.4 \uparrow 0.7$ |
| ENT | ✓ | ✓ | $\mathbf{44.0 \pm 0.2} \uparrow 3.4$ | $\mathbf{60.5 \pm 0.3} \uparrow 2.6$ | $\mathbf{65.5 \pm 0.1} \uparrow 2.2$ | $75.3 \pm 0.1 \uparrow 0.9$ |

we label **DINO*** and **MSN***, and we use them as the base models for our experiments with $(h_\psi, \mu)$-ensembles and weighted ensemble losses. Appx. B.2.1 describes the details for getting such strong performance for DINO* and MSN*. In particular, we find that the projection head has a crucial impact on label efficiency of representations and using a smaller head (3-layer MLP with hidden size 1024) significantly improves few-shot evaluation performance (see Appx. C.2).

**Evaluation metrics** We compared models trained with and without our $(h_\psi, \mu)$-ensembles by measuring various evaluation metrics on ImageNet-1K (Deng et al., 2009). The evaluation metrics reflect the *decodability* and the *label efficiency* of learned representations. We measured the *decodability* with respect to both the linear classifier following the common linear evaluation protocol and the $k$-NN classifier following Caron et al. (2021). We measured the *label efficiency* by evaluating the linear evaluation performance in few-shot settings, including 1% (~13-shots) labeled data evaluation (Chen et al., 2020a) and 1-/2-/5-shot evaluations (Assran et al., 2022). All evaluations used frozen representations of the teacher encoder – we did not fine tune the models. See Appx. B.3 for details.

## 5.1 EMPIRICAL STUDY OF $(h_\psi, \mu)$-ENSEMBLES

Table 1 compares different strategies for **where** and **how to ensemble**. Fig. 4 compares the impact of the weighted ensemble loss on $(h_\psi, \mu)$-**ensemble diversity**. Fig. 3 shows the effect of **increasing the number of ensembles**, **adjusting the temperature** $\gamma$, and **increasing baseline projection head parameters**. In these experiments, we used DINO* with ViT-S/16 trained for 300 epochs as the base model. We compared different ensemble methods applied to the base model with $m = 16$ heads which we found to work the best. For the ENT strategy in Table 1, the entropy weighting temperature $\gamma$ is set to $0.05 \times \log(c)$ by default which is selected from $\{0.0125, 0.025, 0.05, 0.1, 0.2\} \times \log(c)$, where the scale $\log(c)$ gives the maximum entropy of the codebook size $c$. For PROB, we keep $\gamma = 1$.

**Where to ensemble** We study the **where** question by ensembling either the projection head $h_\psi$, the codebook $\mu$, or both with the ENT and the PROB ensemble strategies, as shown in Table 1. We find that ensembling both $h_\psi$ and $\mu$ provides the largest gains for both losses, probably due to the increased flexibility for learning a diverse ensemble. Interestingly, only ensembling $h_\psi$ also works well for the ENT strategy.

**How to ensemble** We study the **how** question by considering four different loss variants: UNIF, PROB, ENT, and the variant of ENT with student entropy weighting. We find that when we ensemble both the projection head $h_\psi$ and the codebook $\mu$, the ENT ensemble strategy leads to the most significant gains (e.g., 3.4 p.p. gains for 1-shot and 0.9 p.p. gains for full-data). The PROB strategy also consistently improves the performance with a slightly larger gain (1 p.p.) in full-data evaluation. In contrast, we see no gains for the UNIF strategy over the baseline. We also study a variant of ENT that uses the student entropy (i.e., Eq. (12) with the $\log \delta(i - j)$ term) for the importance weights (denoted as ENT-ST). ENT-ST performs much worse than ENT and is even worse than the baseline.

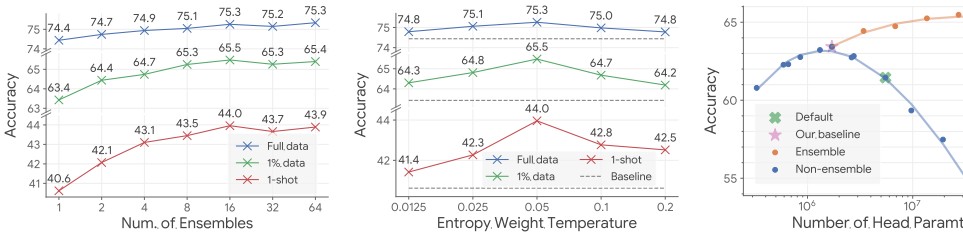

(a) Scaling of $(h_\psi, \mu)$-ensembles.    (b) Effect of ENT temperature $\gamma$.    (c) Comparing different heads.

Figure 3: **Empirical study of $(h_\psi, \mu)$-ensembles.** (a) The gains of $(h_\psi, \mu)$-ensembles start to diminish above 16 heads. (b) The temperature for entropy weighting has a larger impact on few-shot performance. 16 heads are used and $\gamma$ is scaled by $\log(c)$. (c) Our $(h_\psi, \mu)$-ensembles outperform all non-ensembled baselines when controlling for number of parameters. A too powerful non-ensembled projection head significantly harms accuracy. 1%-data evaluation is shown. Also see Fig. 5.

We conjecture that this is because the student predictions typically have a larger variance than teacher predictions (Wang et al., 2022) especially when multi-crop augmentation (Caron et al., 2020; 2021) is applied to the student. Similar experiments on Eq. (11) and/or $\gamma = 0$ variants of PROB also resulted in inferior performance (see Table 12).

**Analysis of $(h_\psi, \mu)$-ensemble diversity**    The previous experiments showed that the choice of ensemble weighting strategy has a large impact on performance. We hypothesize that this choice substantially impacts the diversity of the codebook ensembles. Since the codes in different heads may not be aligned, we align them by the similarity of their code assignment probabilities across different input images, which measures how the codes are effectively used to 'cluster' the data. See Appx. C.4 for detailed explanations and results. In Fig. 4, we visualize the decay patterns of the similarity score between aligned codes (1.0 means the most similar) in a random pair of heads for each weighting strategy. ENT decays the fastest and UNIF decays the slowest, indicating that ENT learns the most diverse codebooks while UNIF is least diverse. This shows a positive correlation between the diversity of $(h_\psi, \mu)$-ensembles and the empirical

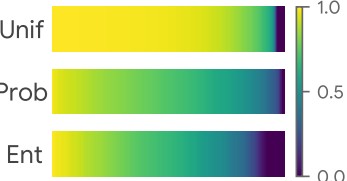

Figure 4: **Visualization of code similarity.** ENT learns the most diverse $(h_\psi, \mu)$-ensembles reflected by the fastest decay of similarity scores between aligned codes in different heads. UNIF has low diversity between heads.

performance of the ensemble strategies from Table 1. Finally, for UNIF, we find that different heads tend to learn the same semantic mappings even when randomly initialized; i.e., the code assignments in different heads become homogeneous up to permutation. See Fig. 8 for a visualization.

**Number of $(h_\psi, \mu)$-ensembles**    We study the effect of increasing the number of $(h_\psi, \mu)$-ensembles $m$ for ENT in Fig. 3a. Having more $(h_\psi, \mu)$-ensembles boosts the performance until $m = 16$. Interestingly, using as few as $m = 2$ heads already significantly improves over the baseline.

**Effect of ENT temperature $\gamma$**    Fig. 3b studies the effect of entropy weighting temperature $\gamma$ for different evaluation metrics. We observe that $\gamma$ has a relatively larger impact on few-shot evaluation performance. $\gamma$ should be neither too high nor too low: a high temperature leads to under-specialization (i.e. less diversity) of heads similar to UNIF ($\gamma \to \infty$) and a low temperature may otherwise lead to over-specialization (i.e., only a single head is used for each input).

**Comparison of different projection heads**    Our method linearly increases projection head parameters, thus a natural question is: Is the gain of $(h_\psi, \mu)$-ensembles due to the increased power (or number of parameters) in projection heads? We answer this question with an empirical study of non-ensembled projection heads. Specifically, we studied non-ensembled $h_\psi$ with (depth, width) searched over $\{2, 3, 4\} \times \{512, 1024, 2048, 4096\}$ and measured the linear evaluation performance with different amounts of labeled data. In Fig. 3c, we plot the 1%-data evaluation result with respect to the number of parameters of the projection head both for ensembled and non-ensembled baselines. See Appx. C.2 for detailed analysis and extra results for other metrics. Our key findings are:

Table 2: **Effectiveness of ensemble heads** for DINO*/MSN* with different ViT models. Our ensemble heads consistently improve all downstream evaluation metrics on ImageNet-1K and achieve a new state-of-the-art for few-shot evaluations. For ViT-S/16, we report linear evaluation results probed from the last layer (left) and from the last 4 layers (right, following DINO). †We evaluated the few-shot settings using DINO's publicly-available pretrained weights in the cases those results were not reported in Caron et al. (2021). ‡MSN ViT-B/16 and ViT-B/8 are both trained for 600 epochs in Assran et al. (2022), whereas our models are trained for only 400, 300 epochs, respectively. For each architecture, we highlight the best DINO baseline and weighted ensemble in blue . For MSN, the corresponding highlights are yellow . The best results for each architecture and metric are **bolded**.

| Method | Few-shot | | | | Full-data | |
|---|---|---|---|---|---|---|
| | 1 | 2 | 5 | ~13 (1%) | $k$-NN | Linear |
| *ViT-S/16, 800 epochs* | | | | | | |
| iBOT | $40.4 \pm 0.5$ | $50.8 \pm 0.8$ | $59.9 \pm 0.2$ | 65.9 | **75.2** | - / **77.9** |
| DINO | $38.9 \pm 0.4$ | $48.9 \pm 0.3$ | $58.5 \pm 0.1$ | 64.5 | 74.5 | 76.1 / 77.0 |
| DINO (Repro) | $39.1 \pm 0.3$ | $49.1 \pm 0.5$ | $58.6 \pm 0.2$ | 64.7 | 74.3 | 75.8 / 76.9 |
| DINO* (Retuned) | $44.6 \pm 0.2$ | $53.6 \pm 0.3$ | $61.1 \pm 0.2$ | 66.2 | 74.1 | 75.8 / 76.9 |
| MSN | $47.1 \pm 0.1$ | $55.8 \pm 0.6$ | $62.8 \pm 0.3$ | 67.2 | - | - / 76.9 |
| MSN* (Retuned) | $47.4 \pm 0.1$ | $56.3 \pm 0.4$ | $62.8 \pm 0.2$ | 67.1 | 73.3 | 75.6 / 76.6 |
| DINO*-PROB (16) | $45.2 \pm 0.4$ | $54.9 \pm 0.4$ | $62.5 \pm 0.2$ | 67.3 | 75.1 | 76.5 / 77.6 |
| DINO*-ENT (4) | $46.3 \pm 0.1$ | $55.5 \pm 0.6$ | $63.0 \pm 0.3$ | 67.5 | 74.8 | 76.2 / 77.2 |
| DINO*-ENT (16) | $47.6 \pm 0.1$ ↑ **3.0** | $56.8 \pm 0.5$ | $64.0 \pm 0.2$ | 68.3 ↑ **2.1** | 75.3 | **76.8 / 77.7** ↑ **0.8** |
| MSN*-ENT (2) | $48.8 \pm 0.2$ | $57.5 \pm 0.5$ | $64.0 \pm 0.2$ | 67.9 | 74.6 | 76.0 / 76.9 |
| MSN*-ENT (8) | **$50.1 \pm 0.1$** ↑ **2.7** | **$58.9 \pm 0.6$** | **$65.1 \pm 0.3$** | **68.7** ↑ **1.6** | 75.2 | 76.4 / 77.4 ↑ **0.8** |
| *ViT-B/16, 400 epochs* | | | | | | |
| iBOT | $46.1 \pm 0.3$ | $56.2 \pm 0.7$ | $64.7 \pm 0.3$ | 69.7 | **77.1** | **79.5** |
| DINO† | $43.0 \pm 0.2$ | $52.7 \pm 0.5$ | $61.8 \pm 0.2$ | 67.4 | 76.1 | 78.2 |
| DINO* (Retuned) | $49.3 \pm 0.1$ | $58.1 \pm 0.5$ | $65.0 \pm 0.3$ | 69.1 | 76.0 | 78.5 |
| MSN‡ | $49.8 \pm 0.2$ | $58.9 \pm 0.4$ | $65.5 \pm 0.3$ | - | - | - |
| MSN* (Retuned) | $50.7 \pm 0.1$ | $59.2 \pm 0.4$ | $65.9 \pm 0.2$ | 69.7 | 74.7 | 78.1 |
| DINO*-ENT (16) | $52.8 \pm 0.1$ ↑ **3.5** | $61.5 \pm 0.4$ | $67.6 \pm 0.3$ | 71.1 ↑ **2.0** | **77.1** | 79.1 ↑ **0.6** |
| MSN*-ENT (8) | **$53.7 \pm 0.2$** ↑ **3.0** | **$62.4 \pm 0.6$** | **$68.3 \pm 0.2$** | **71.5** ↑ **1.8** | **77.2** | 78.9 ↑ **0.8** |
| *ViT-B/8, 300 epochs* | | | | | | |
| DINO† | $47.5 \pm 0.2$ | $57.3 \pm 0.5$ | $65.4 \pm 0.3$ | 70.3 | 77.4 | 80.1 |
| DINO* (Retuned) | $49.5 \pm 0.5$ | $58.6 \pm 0.6$ | $65.9 \pm 0.3$ | 70.7 | 77.1 | 80.2 |
| MSN‡ | $55.1 \pm 0.1$ | $64.9 \pm 0.7$ | $71.6 \pm 0.3$ | - | - | - |
| MSN* (Retuned) | $51.9 \pm 0.3$ | $61.1 \pm 0.4$ | $67.7 \pm 0.3$ | 71.7 | 75.7 | 80.3 |
| DINO*-ENT (16) | $55.0 \pm 0.4$ ↑ **5.5** | $63.4 \pm 0.6$ | $69.5 \pm 0.3$ | 73.4 ↑ **2.7** | 78.6 | **81.0** ↑ **0.8** |
| MSN*-ENT (8) | **$55.6 \pm 0.2$** ↑ **3.7** | **$64.5 \pm 0.5$** | **$70.3 \pm 0.2$** | **73.4** ↑ **1.7** | **78.9** | 80.8 ↑ **0.5** |

- A too powerful non-ensembled $h_\psi$ significantly hurts the label efficiency of learned representations. This result is similar to Chen et al. (2020b), which found that probing from intermediate layers of projection heads (which can be viewed as using a shallower head) could improve semi-supervised learning (1%-/10% labeled data) results.

- The default head (3/2048, denoted as 'Default') used in recent SSL methods (SimCLRv2, DINO, MSN, etc.) does not perform as well in few-shot evaluations, probably because it is selected by looking at full-data evaluation metrics. In contrast, our baseline (3/1024, denoted as 'Our baseline') significantly improves few-shot evaluation performance.

- Our $(h_\psi, \mu)$-ensembles outperform all non-ensembled baselines and lead to consistent improvements in all evaluation metrics, despite the increase of parameters.

## 5.2 IMPROVING SOTA RESULTS WITH ENSEMBLEING

Next we apply $(h_\psi, \mu)$-ensembles to DINO* and MSN* and compare with the state-of-the-art results. We experimented with model architectures ViT-S/16, ViT-B/16, ViT-B/8 trained for 800, 400, 300 epochs respectively following Caron et al. (2021). We include both the published results and our retuned versions to ensure strong baselines. For clarity, we denote our method as "{baseline}-{ensemble strategy} (# of heads)", e.g., DINO*-ENT (4). We tuned both baselines and our methods for all architectures. We report the best hyperparameters for all models in Appx. B.2.2.

Table 2 compares the results of $(h_\psi, \mu)$-ensembles and baselines. We find that $(h_\psi, \mu)$-ensembles with ENT consistently improve *all evaluation metrics* (full-data, few-shot) across *both SSL methods* (DINO*, MSN*) and *all architectures* (ViT-S/16, ViT-B/16, ViT-B/8) over their non-ensembled counterparts. The gains in few-shot evaluation are particularly substantial, providing a new state-of-the-art for ImageNet-1K evaluation from ImageNet pretraining.

## 5.3 More evaluations for $(h_\psi, \mu)$-ensembles

Table 3: **Comparison of transfer performance.** ViT-S/16 is used. Our ensemble heads lead to consistent improvements for MSN* and comparable results for DINO*.

|  | Food101 | CIFAR10 | CIFAR100 | SUN397 | Cars | DTD | Pets | Caltech-101 | Flowers | Avg. |
|---|---|---|---|---|---|---|---|---|---|---|
| DINO* | 78.4 | 93.8 | 81.0 | 66.1 | 66.7 | 74.6 | 92.0 | **94.9** | **94.4** | 82.43 |
| DINO*-ENT (16) | **79.1** | 93.8 | **81.4** | **66.5** | 66.8 | **74.9** | **92.8** | 94.6 | 93.9 | 82.64 |
| MSN* | 77.7 | 93.1 | 79.8 | 64.6 | 63.3 | 72.2 | 92.4 | 94.7 | 92.7 | 81.17 |
| MSN*-ENT (8) | **78.4** | **93.9** | **81.1** | **65.2** | **68.0** | **73.2** | **93.1** | **95.4** | **92.8** | 82.34 |

**Transfer learning** In Table 3, we compare the transfer learning performance of $(h_\psi, \mu)$-ensembles and non-ensembled baselines. We used ViT-S-16 models trained on ImageNet-1K for 800 epochs and evaluated on 9 natural downstream datasets from Chen et al. (2020a) with linear evaluation (details in Appx. B.3). $(h_\psi, \mu)$-ensembles lead to consistent improvements in transfer performance for MSN* and comparable results for DINO*.

**Training overhead** In Table 4, we benchmark the computational overhead of $(h_\psi, \mu)$-ensembles at training time. We used a medium sized model, DINO* with ViT-B/16, trained with the same setting used in all of our experiments. We benchmarked the wall-clock time and peak memory on 128 TPUv3 cores. $(h_\psi, \mu)$-ensembling is relatively cheap in terms of training cost because the ensembled parts typically account for a small portion of total computation, especially when the backbone encoder is more computationally expensive (e.g., ViT-B/8). Again, we emphasize that there is no evaluation overhead when $(h_\psi, \mu)$-ensembles are removed.

Table 4: **Training overhead.** Wall-clock time and peak memory per core for training with different numbers of ensembles.

| m | Wall Time | Peak Memory |
|---|---|---|
| 1 | 5.81h | 5.25G |
| 4 | 5.91h | 5.40G |
| 16 | 6.34h | 5.89G |

## 6 Conclusion & Discussion

We introduced an efficient ensemble method for SSL where multiple projection heads are ensembled to effectively improve representation learning. We showed that with carefully designed ensemble losses that induce diversity over ensemble heads, our method significantly improves recent state-of-the-art SSL methods in various evaluation metrics, particularly for few-shot evaluation. Although ensembling is a well-known technique for improving evaluation performance of a single model, we demonstrated that, for models with throw-away parts such as the projection heads in SSL, ensembling these parts can improve the learning of the non-ensembled representation encoder and also achieve significant gains in downstream evaluation without introducing extra evaluation cost.

Our ensemble method is applicable to many SSL methods beyond the two we explored. For example, one may consider generalization to BYOL (Grill et al., 2020) or SimSiam (Chen & He, 2021) that ensembles projection and/or prediction heads, or MAE (He et al., 2022) that ensembles the decoders (which introduces more training cost though). Our weighted ensemble losses can also be applied as long as the original loss can be reformulated as MLE for some $t$, $s$, and $Y$, e.g., the MSE loss in these methods is MLE under multivariate normal distributions. We hope our results and insights will motivate more future work for extending our method or exploring more ensemble techniques for SSL.

In future work, we also hope to remove three limitations of our setting. First, considering ensembling strategies that include the representation encoder, $r_\omega$, may lead to further improvements in the performance of weighted ensemble SSL, at the cost of increased computation requirements during both training and evaluation. Second, considering heterogenous architectures in the ensemble may further improve the learned representations (e.g., mixing Transformers with ResNets), whether the heterogeneity is in $r_\omega$, $h_\psi$, or both. Third, considering other possibilities for $f_{ijy}$ may also reveal performance gains and improve our understanding of the critical aspects that lead to good SSL representations, similar to what we learned about the importance of ensemble diversity.

ACKNOWLEDGMENTS

We would like to thank Mathilde Caron and Mahmoud Assran for their extensive help in reproducing DINO and MSN baselines. We would also like to thank Ting Chen and Yann Dubois for their helpful discussions and encouragements.

REPRODUCIBITLITY STATEMENT

We include detailed derivations for all our proposed losses in Appx. D. We report experimental details in Appx. B, including the implementation details for reproducing the baselines (Appx. B.1), training and evaluating our methods (Appx. B.2.1), and all hyper-parameters (Appx. B.2.2) used in our experiments for reproducing our results in Table 2.

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

# A  PSEUDOCODE

---

**Algorithm 1:** Pseudocode for computing ensemble loss

```
# b, n, c,:  batch size, number of ensemble heads, codebook size
# log_ps, log_pt:  student, teacher log probabilities with n ensembles
# strategy:  ensemble loss average strategy
# tau_ent:  temperature for entropy weighting

def ensemble_loss(log_ps, log_pt, strategy, tau_ent):
    b, n, c = log_pt.shape # axis 1 corresponds to ensemble
    log_pt = stop_grad(log_pt) # stop gradient for teacher

    if strategy == "Unif":
        loss = - (exp(log_pt) * log_ps).sum(axis=-1)
        loss = loss.mean(axis=1) # average over ensembles
    elif strategy == "Prob":
        log_mean_pt = logsumexp(log_pt, axis=1, b=1/n) # mean teacher
        log_mean_ps = logsumexp(log_ps, axis=1, b=1/n) # mean student
        loss = - (exp(log_mean_pt) * log_mean_ps).sum(axis=-1)
    elif strategy == "Ent":
        ent = - (exp(log_pt) * log_pt).sum(axis=-1) # teacher entropy
        weight = softmax(-ent/tau_ent, axis=1) # entropy weights
        loss = - (exp(log_pt) * log_ps).sum(axis=-1)
        loss = (loss * weight).sum(axis=1) # entropy weighted average

    return loss.mean() # average over samples
```

---

**Algorithm 2:** Pseudocode for ensemble heads with simplified DINO

```
# n, c, eta:  number of ensemble heads, codebook size, momentum update rate
# fs, ft:  student, teacher encoders
# hs_ens, ht_ens:  student, teacher projection heads with n ensembles, list with length n
# mus_ens, mut_ens:  student, teacher codebooks with n ensembles, list with length n
# taus, taut:  student, teacher temperatures
# strategy:  ensemble loss average strategy
# tau_ent:  temperature for entropy weighting

for x in dataloader:  # load a batch with b samples
    xs, xt = augs(x), augt(x) # random augmentations
    zs, zt = fs(xs), ft(xt) # representations, (b, l)

    # all following computation can be parallelized with batch computation
    log_ps, log_pt = [], []
    for j in range(n):
        hs_j, ht_j = hs_ens[j], ht_ens[j] # j-th projection head
        mus_j, mut_j = mus_ens[j], mut_ens[j] # j-th codebook, (d, c)

        es_j, et_j = hs_j(zs), ht_j(zt) # j-th embedding, (b, d)

        rs_j = (es_j @ mus_j) / (es_j.norm(axis=1, keepdims=True) * mus_j.norm(axis=0,
         keepdims=True)) / taus # student logits, (b, c)
        rt_j = (et_j @ mut_j) / (et_j.norm(axis=1, keepdims=True) * mut_j.norm(axis=0,
         keepdims=True)) / taut # teacher logits, (b, c)

        log_ps_j = logsoftmax(rs_j, axis=-1) # (b, c)
        log_pt_j = logsoftmax(rt_j, axis=-1) # (b, c)
        log_pt_j = renorm(log_pt_j) # adjust teacher predictions with centering or sinkhorn,
         omitted here for simplicity

        log_ps.append(log_ps_j)
        log_pt.append(log_pt_j)

    log_ps = stack(log_ps_j, axis=1) # stacked student log probablities, (b, n, c)
    log_pt = stack(log_pt_j, axis=1) # stacked teacher log probablities, (b, n, c)

    loss = ensemble_loss(log_ps, log_pt, strategy=strategy) # compute ensemble loss

    loss.backward() # back-propagate
    sgd_update(fs, hs_ens, mus_ens) # apply gradient decent update for student
    ema_update(ft, ht_ens, mut_ens, rate=eta) # apply momentum update for teacher
```

---

## B EXPERIMENTAL DETAILS

In this section, we provide details for our experiments. In Appx. B.1, we describe how we reproduced and improved the baseline DINO/MSN models. We give the implementation details for SSL training and evaluation in Appx. B.2 and Appx. B.3 respectively. All the hyper-parameters used in our experiments are in Appx. B.2.2.

### B.1 REPRODUCING & IMPROVING BASELINES

We carefully reproduced and further improved baseline methods (denoted as DINO* and MSN* respectively) with an extensive study and hyperparameter search (see Appx. B.1). In particular, we systematically study the projection head design (which we found is crucial for few-shot evaluation performance (Appx. C.2)) and different techniques for avoiding collapse used in both methods (Appx. C.1). DINO* performs significantly better than DINO on few-shot evaluation (e.g., 2~6 percentage point (p.p.) gains for 1 shot) and maintains the full-data evaluation performance. The main adjustments of DINO* are: (i) A 3-layer projection head with a hidden dimension of 1024 (instead of 2048); (ii) Sinkhorn–Knopp (SK) normalization (instead of centering) is applied to teacher predictions, combined with a smaller teacher temperature $\tau = 0.025$ and codebook size $c =$1024 or 4096. MSN* uses the same projection head as DINO* and applies ME-MAX regularization without SK normalization (which is applied in MSN by default). Further details for DINO and MSN can be found below.

#### B.1.1 DINO

Table 5: **Reproducing & Improving DINO.** Our reproduce results match the public numbers. We further improve the DINO baseline (DINO*) by studying projection heads and collapse-avoiding techniques. The evaluation results of DINO/DINO* ViT-S/16 trained with 800 epochs are reported.

| | **Few-shot** | | | | **Full-data** | |
|---|---|---|---|---|---|---|
| | 1 | 2 | 5 | ~13 (1%) | $k$-NN | Linear |
| DINO (Caron et al., 2021) | $38.9 \pm 0.4$ | $48.9 \pm 0.3$ | $58.5 \pm 0.1$ | 64.5 | 74.5 | 76.1 / 77.0 |
| DINO (Ours reproduced) | $39.1 \pm 0.3$ | $49.1 \pm 0.5$ | $58.6 \pm 0.2$ | 64.7 | 74.3 | 75.8 / 76.9 |
| DINO* (Retuned) | $44.6 \pm 0.2$ | $53.6 \pm 0.3$ | $61.1 \pm 0.2$ | 66.2 | 74.1 | 75.8 / 76.9 |

**Reproducing DINO** We carefully reproduced DINO with JAX following the official DINO implementation[1]. In Table 5, we report the evaluation results of DINO using ViT-S trained with 800 epochs following the exact training configuration for ViT-S/16 in the official DINO code. The official results of full-data evaluation and 1%-data evaluation are from Caron et al. (2021), the other few-shot evaluation results are evaluated by Assran et al. (2022) and also validated by us. Note that for consistency of full-data linear evaluation, we report the results with both the `[CLS]` token representations of the last layer and the concatenation of the `[CLS]` token representations from the last 4 layers following Caron et al. (2021). For 1-/2-/5-shots evaluation results, we report the mean accuracy and standard deviation across 3 random splits of the data following Assran et al. (2022). As shown in Table 5, our reproduced results are all comparable with the published numbers which validates the implementation of our training and evaluation pipelines.

**Improving DINO** We improved the DINO baseline with a systematic empirical study of some important components. We first empirically compared different techniques for avoiding collapse (see Appx. C.1) and find that Sinkhorn-Knopp (SK) normalization is a more effective and also simpler technique for encouraging codebook usage than the centering operation used in DINO. We thus applied SK normalization, which enabled us to use a smaller teacher temperature $\tau = 0.025$ (instead of $\tau = 0.07$) and a much smaller codebook size $c =$1024 or 4096 (instead of 65536). These modifications lead to similar performance as DINO with a much smaller codebook (up to 1M parameters, compared to 16M parameters for DINO). Next we empirically studied the effect of projection heads for different evaluation metrics (see Appx. C.2), and found that the design of

---

[1]`https://github.com/facebookresearch/dino`

projection heads is crucial for few-shot evaluation metrics and an too power powerful projection head (e.g., the 3-layer MLP with a hidden dimension of 2048 used in DINO/MSN/etc.) could significantly hurt the few-shot performance. With an empirically study of projection head architectures, we found that a simply reducing the hidden dimension to 1024 could significantly improves the few-shot evaluation performance while maintaining full-data evaluation performance. The improved results of DINO* are shown in Table 5.

### B.1.2   MSN

Table 6: **Reproducing & improving MSN.** We implement MSN* by adding ME-MAX regularization and masking to DINO*, which surpasses public MSN results. The evaluation results of MSN/MSN* ViT-S/16 trained with 800 epochs are reported.

| | Few-shot | | | | Full-data | |
|---|---|---|---|---|---|---|
| | 1 | 2 | 5 | $\sim$13 (1%) | $k$-NN | Linear |
| MSN (Assran et al., 2022) | $47.1 \pm 0.1$ | $55.8 \pm 0.6$ | $62.8 \pm 0.3$ | 67.2 | - | -  / 76.9 |
| MSN (Repro) | $39.1 \pm 0.3$ | $49.2 \pm 0.3$ | $58.4 \pm 0.1$ | 64.3 | 72.8 | 74.7 / 75.5 |
| MSN* (Retuned) | $47.4 \pm 0.1$ | $56.3 \pm 0.4$ | $62.8 \pm 0.2$ | 67.1 | 73.3 | 75.6 / 76.6 |

We carefully implemented MSN by adding its main components, i.e., ME-MAX regularization and masking, to the DINO implementation (denoted as MSN*), which surpassed public results as shown in Table 6. Note that the implementation of MSN* does not exactly match the public implementation in the public MSN code[2], where the main differences are:

- MSN applies ME-MAX with Sinkhorn-Knopp normalization by default (as in the released training configuration), which we empirically find does not work very well (see Table 9). MSN* does not apply SK normalization and tunes the regularization strength for ME-MAX.

- Some differences in implementation details, e.g., schedules for learning rate/weight decay, batch normalization in projection heads, specific data augmentations, etc. MSN* uses the exact same setup as DINO* which follows original DINO implementation.

We initially tried to exactly reproduce the original MSN following the public MSN code, but the results are much below the public ones, as shown in Table 6. Incorporating the two differences above bridges the gap and makes MSN* surpass the public results.

### B.2   PRETRAINING DETAILS

In this subsection, we provide the general implementation details in Appx. B.2.1 and specific hyperparameters in Appx. B.2.2 in Appx. B.2.2 for reproducibility.

### B.2.1   IMPLEMENTATION DETAILS

**Common setup**   We experimented with DINO (Caron et al., 2021) and MSN (Assran et al., 2022) models on ImageNet ILSVRC-2012 dataset (Deng et al., 2009). We mainly followed the training setup in Caron et al. (2021). In particular, all models were trained with AdamW optimizer (Loshchilov & Hutter, 2018) and a batch size of 1024. The learning rate was linearly warmuped to 0.002 ($=0.001\times$batch size/512) and followed a cosine decay schedule. The weight decay followed a cosine schedule from 0.04 to 0.4. The momentum rate for the teacher was increased from 0.996 to 1 with a cosine schedule following BYOL (Grill et al., 2020). A stochastic depth (Huang et al., 2016) of 0.1 was applied without dropout (Srivastava et al., 2014). The student temperature $\tau$ is set to 0.1. As with DINO, we used the data augmentations of BYOL and multi-crop augmentation of SWAV (Caron et al., 2020). In particular, 2 global views with a 224$\times$224 resolution and crop area range [0.25, 1.0] were generated for the teacher and student, and another 10 local views with 96$\times$96 resolution and crop area range [0.08, 0.25] were used as extra augmented inputs for the student. For MSN, we used the exact same setup and incorporated its major component: 1) mean entropy maximization (ME-MAX) regularization; 2) masking as an extra augmentation applied to the student global view.

---

[2]https://github.com/facebookresearch/msn

**Main modifications**    We retuned the baselines (DINO* and MSN*) as detailed in Appx. B.1, and the main adjustments are as followed. We used a 3-layer projection head with a hidden dimension of 1024. The output embedding (i.e., $(h_\psi \circ r_\omega)(x)$) and the codes (i.e., $\mu$) both have a dimension of 256 and are $L_2$ normalized. For DINO*, Sinkhorn–Knopp (SK) normalization was applied to teacher predictions. For MSN*, ME-MAX was used without SK normalization and the regularization strength was tuned over $\{3, 4, 5\}$. For all models, we used teacher temperature $\tau = 0.025$ which was linearly decayed from 0.05 for the first 30 epochs. The codebook size $c$ was selected over $\{1024, 4096\}$ for all models, and typically $c = 4096$ was selected for baseline methods and $c = 1024$ was selected for ours. For our $(h_\psi, \mu)$-ensembles with ENT, entropy weighting temperature $\gamma$ is linearly decayed from 0.5 to the specified value.

### B.2.2    HYPER-PARAMETERS

We report the hyperparameters for training our models for reproducibility:

Table 7: Hyper-parameters for training the DINO* model.

| Hyper-parameter | ViT-S/16 | | | ViT-B/16 | | ViT-B/8 | |
|---|---|---|---|---|---|---|---|
| | DINO* | DINO*-PROB (16) | DINO*-ENT (4/16) | DINO* | DINO*-ENT (16) | DINO* | DINO*-ENT (16) |
| training epoch | | 800 | | | 400 | | 300 |
| batch size | | 1024 | | | 1024 | | 1024 |
| learning rate | | 2e-3 | | | 2e-3 | | 2e-3 |
| warmup epoch | | 10 | | | 30 | | 10 |
| min lr | | 1e-5 | | | 1e-5 | | 4e-5 |
| weight decay | | 0.04 → 0.4 | | | 0.04 → 0.4 | | 0.04 → 0.4 |
| stochastic depth | | 0.1 | | | 0.1 | | 0.1 |
| gradient clip | | 3.0 | | | 1.0 | | 3.0 |
| momentum | | 0.996 → 1.0 | | | 0.996 → 1.0 | | 0.996 → 1.0 |
| # of multi-crops | | 10 | | | 10 | | 10 |
| masking ratio | | - | | | - | | - |
| proj. layer | | 3 | | | 3 | | 3 |
| proj. hidden dim | | 1024 | | | 1024 | | 1024 |
| emb. dim $d$ | | 256 | | | 256 | | 256 |
| rep. dim | | 384 | | | 768 | | 768 |
| codebook size $c$ | 4096 | 1024 | 1024 | 4096 | 1024 | 4096 | 1024 |
| student temp. | | 0.1 | | | 0.1 | | 0.1 |
| teacher temp. | | 0.025 | | | 0.025 | | 0.025 |
| te. temp. decay epoch | | 30 | | | 30 | | 30 |
| center | | ✗ | | | ✗ | | ✗ |
| SK norm | | ✓ | | | ✓ | | ✓ |
| ME-MAX weight | | - | | | - | | - |
| ent. weight temp. $\gamma$ | - | - | 0.05 | - | 0.05 | - | 0.06 |
| $\gamma$ init. | - | - | 0.5 | - | 0.5 | - | 0.5 |
| $\gamma$ decay epoch | - | - | 30 | - | 30 | - | 30 |

### B.3    EVALUATION PROTOCALS

**Few-shot linear evaluation**    We followed the few-shot evaluation protocal in Assran et al. (2022). Specifically, we used the 1-/2-/5-shot ImageNet dataset splits[3] in Assran et al. (2022) and 1% ($\sim$13-shot) ImageNet dataset splits[4]. For given labelled images, we took a single central crop of size $224 \times 224$ without additional data augmentations, and extracted the output [CLS] token representations from the frozen pretrained model. Then we trained a linear classifier with multi-class logistic regression on top of the extracted representations. We used the scikit-learn package (Pedregosa et al., 2011) for the logistric regression classifier. For all few-shot evaluations, we searched the $L_2$ regularization strength over $\{$1e-4, 3e-4, 1e-3, 3e-3, 1e-2, 3e-2, 1e-1, 3e-1, 1, 3, 10$\}$.

**Full-data linear evaluation**    We followed the linear evaluation protocal in (Caron et al., 2021). Specifically, we trained a linear classifier on top of the representations extracted from the frozen pretrained model. The linear classifier is optimized by SGD with Nesterov momentum (Nesterov, 1983; Sutskever et al., 2013) of 0.9 and a batch size of 4096 for 100 epochs on the whole ImageNet dataset, following a cosine learning rate decay schedule. We did not apply any weight decay.

---

[3]Publicly available at https://github.com/facebookresearch/msn
[4]Publicly available at https://github.com/google-research/simclr/tree/master/imagenet_subsets

Table 8: Hyper-parameters for training the MSN* model.

| Hyper-parameter | ViT-S/16 | | ViT-B/16 | | ViT-B/8 | |
|---|---|---|---|---|---|---|
| | DINO* | MSN*-ENT (2/8) | MSN* | MSN*-ENT (8) | MSN* | MSN*-ENT (8) |
| training epoch | 800 | | 400 | | 300 | |
| batch size | 1024 | | 1024 | | 1024 | |
| learning rate | 2e-3 | | 2e-3 | | 2e-3 | |
| warmup epoch | 20 | | 30 | | 20 | |
| min lr | 1e-5 | | 4e-5 | | 4e-5 | |
| weight decay | $0.04 \rightarrow 0.4$ | | $0.04 \rightarrow 0.4$ | | $0.04 \rightarrow 0.4$ | |
| stochastic depth | 0.1 | | 0.1 | | 0.1 | |
| gradient clip | 1.0 | | 1.0 | | 1.0 | |
| momentum | $0.996 \rightarrow 1.0$ | | $0.996 \rightarrow 1.0$ | | $0.996 \rightarrow 1.0$ | |
| # of multi-crops | 10 | | 10 | | 10 | |
| masking ratio | 0.2 | | 0.2 | | 0.15 | |
| proj. layer | 3 | | 3 | | 3 | |
| proj. hidden dim | 1024 | | 1024 | | 1024 | |
| emb. dim $d$ | 256 | | 256 | | 256 | |
| rep. dim | 384 | | 768 | | 768 | |
| codebook size $c$ | 4096 | 1024 | 4096 | 1024 | 4096 | 1024 |
| student temp. | 0.1 | | 0.1 | | 0.1 | |
| teacher temp. | 0.025 | | 0.025 | | 0.025 | |
| te. temp. decay epoch | 30 | | 30 | | 30 | |
| center | ✗ | | ✗ | | ✗ | |
| SK norm | ✗ | | ✗ | | ✗ | |
| ME-MAX weight | 4.0 | | 4.0 | | 4.0 | |
| ent. weight temp. $\gamma$ | - | 0.01 | - | 0.005 | - | 0.01 |
| $\gamma$ init. | - | 0.5 | - | 0.5 | - | 0.5 |
| $\gamma$ decay epoch | - | 30 | - | 30 | - | 30 |

During training, we only applied basic data augmentations including random resized crops of size $224 \times 224$ and horizontal flips. During testing, we took a single central crop of the same size. For ViT-S/16, Caron et al. (2021) found that concatenating the `[CLS]` token representations from the last $l$ (specifically, $l = 4$) layers (c.f. Appendix F.2 in Caron et al. (2021)) improved the results by about 1 p.p. We followed the same procedure, but reported linear evaluation results with both $l = 1$ and $l = 4$ in Table 2 for consistency. In our empirical study with ViT-S/16, we used the result with $l = 1$. For larger models (e.g., ViT-B/16), we followed Caron et al. (2021); Zhou et al. (2022) to use the concatenation of the `[CLS]` token representation and the average-pooled patch tokens from the last $l = 1$ layer for linear evaluation. For all linear evaluations, we searched the base learning rate over {4.8e-3, 1.6e-2, 4.8e-2, 1.6e-1, 4.8e-1, 1.6}.

**Full-data $k$-NN evaluation** We followed the $k$-NN evaluation protocal in Caron et al. (2021); Wu et al. (2018). Specifically, for each image in the given dataset, we took a single central crop of size $224 \times 224$ without additional data augmentations, and extracted the output `[CLS]` token representations from the frozen pretrained model. The extracted representations are used for a weighted $k$-Nearest-Neighbor classifier. In particular, denote the stored training representations and labels as $\mathcal{D} = \{(z_i, y_i)\}_{i=1}^{N}$. For a test image with extracted representation $z$, denote the set of its top $k$-NN training samples as $\mathcal{D}_k[z] \subseteq \mathcal{D}$ and $|\mathcal{D}_k[z]| = k$. The $k$-NN set $\mathcal{D}_k[z]$ is used to make the prediction for the test image with a weighted vote, i.e., $\hat{y} = \arg\max_y \left( \sum_{(z_j, y_j) \in \mathcal{D}_k[z]} \alpha_j \mathbf{1}_{y=y_j} \right)$, where $\mathbf{1}_{y=y_j}$ is the one-hot vector corresponding to label $y_j$ and $\alpha_j$ is the weight induced by the cosine similarity between $z$ and $z_j$, i.e., $\alpha_j = \exp\left( \frac{1}{\tau'} \frac{z^\mathsf{T} z_j}{\|z\| \|z_j\|} \right)$. We set $\tau' = 0.07$ without tuning as in Caron et al. (2021); Wu et al. (2018). For all $k$-NN evaluations, we searched $k$ over {5, 10, 20, 50, 100} and found that $k = 10$ or $k = 20$ was consistently the best.

**Transfer evaluation via linear probing** We mainly followed the transfer evaluation protocol in (Grill et al., 2020; Chen et al., 2020a). In particular, we used 9 of their 13 datasets that are available in `tensorflow-datasets` (tfd), namely Food-101 (Bossard et al., 2014), CIFAR10 (Krizhevsky et al., 2009), CIFAR100 (Krizhevsky et al., 2009), SUN397 scene dataset (Xiao et al., 2010), Stanford Cars (Krause et al., 2013), Describable Textures Dataset (Cimpoi et al., 2014, DTD), Oxford-IIIT Pets (Parkhi et al., 2012), Caltech-101 (Fei-Fei et al., 2004), Oxford 102 Flowers (Nilsback & Zisserman,

Table 9: **Empirical study of different techniques for avoiding collapse.** Using Sinkhorn-Knopp normalization instead of centering for DINO leads to improved performance, and matches the original DINO even with a much smaller codebook. The ME-MAX regularization of MSN is very effective and leads to significant improvement for few-shot evaluations.

| | Technique | | | Few-shot | | | | Full-data | |
|---|---|---|---|---|---|---|---|---|---|
| | Center | Sinkhorn | ME-MAX | 1 | 2 | 5 | $\sim$13 (1%) | $k$-NN | Linear |
| DINO | ✓ | | | $37.8 \pm 0.4$ | $47.4 \pm 0.3$ | $56.9 \pm 0.4$ | 63.0 | 72.4 | 74.9 |
| | | ✓ | | $39.1 \pm 0.3$ | $49.4 \pm 0.3$ | $58.7 \pm 0.2$ | 64.8 | 74.1 | 76.0 |
| MSN | | ✓ | ✓ | $36.0 \pm 0.4$ | $46.6 \pm 0.6$ | $56.5 \pm 0.2$ | 63.2 | 73.2 | 75.2 |
| | | | ✓ | $43.9 \pm 0.2$ | $53.0 \pm 0.3$ | $61.1 \pm 0.2$ | 66.0 | 74.0 | 75.8 |

Table 10: ME-MAX regularization is sensitive to hyper-parameters.

| Weight | Few-shot | | | | Full-data | |
|---|---|---|---|---|---|---|
| | 1 | 2 | 5 | $\sim$13 (1%) | KNN | Linear |
| 1.0 | $37.6 \pm 0.2$ | $48.0 \pm 0.4$ | $57.7 \pm 0.2$ | 64.0 | 73.5 | 75.6 |
| 3.0 | $43.9 \pm 0.2$ | $53.0 \pm 0.3$ | $61.1 \pm 0.2$ | 66.0 | 74.0 | 75.8 |
| 5.0 | $43.6 \pm 0.2$ | $52.6 \pm 0.4$ | $60.4 \pm 0.1$ | 65.5 | 73.9 | 75.6 |

2008). Following their evaluation metrics, we reported mean per-class accuracy for Oxford-IIIT Pets, Caltech-101, and Oxford 102 Flowers datasets and reported top-1 accuracy for other datasets. We transferred the models pretrained on ImageNet (Deng et al., 2009) to these datasets by training a linear classifer on top of frozen representations. In particular, we resized given images to $256 \times 256$ and took a single central crop of size $224 \times 224$ without additional data augmentations. We extracted the output `[CLS]` token representations from the frozen pretrained model. Then we trained a linear classifier with multi-class logistic regression on top of the extracted representations. We used the `scikit-learn` package (Pedregosa et al., 2011) for the logistric regression classifier. For all transfer evaluations, we searched the $L_2$ regularization strength over {1e-6, 1e-5, 1e-4, 3e-4, 1e-3, 3e-3, 1e-2, 3e-2, 1e-1, 3e-1, 1, 3, 1e1, 3e1, 1e2, 1e3, 1e4, 1e5}.

## C ADDITIONAL RESULTS

### C.1 EMPIRICAL STUDY OF TECHNIQUES FOR AVOIDING COLLAPSE

Most self-supervised learning methods utilize some techniques to avoid collapse of representations with, e.g., contrastive loss (Chen et al., 2020a; He et al., 2020), batch normalization (Grill et al., 2020), asymmetric architecture design with a predictor (Grill et al., 2020; Chen & He, 2021), etc. In DINO and MSN, a learnable codebook is used for the learning objective and different techniques are applied to encourage the effective codebook usage. There are two potential cases of collapse (as discussed in Caron et al. (2021)):

- *Dominating codes*. This is the case of "winner-take-all": only a small portion of codes are being predicted while others are inactive. Typical solutions for avoiding this include applying Sinkhorn–Knopp normalization (Cuturi, 2013) as in SWaV (Caron et al., 2020), centering teacher logits as in DINO (Caron et al., 2021), and applying mean-entropy maximization regularization (ME-MAX) as in MSN (Assran et al., 2022). Note that in MSN, ME-MAX is combined with Sinkhorn–Knopp normalization by default.

- *Uniform codes*. This is the case where all codes are treated equally and the predictions reduce to be uniform over codes. A simple and effective solution is to applying sharpening, i.e., using a lower temperature for computing the teacher prediction.

We systematically study different techniques in a unified setup. In particular, we used DINO with the ViT-S backbone, a 3-layer MLP projection head with hidden dimension 2048, and a codebook of size 4096 and dimension 256. We applied different techniques to DINO and searched the teacher temperature in $\{0.0125, 0.025, 0.05\}$ for each. For ME-MAX, we searched regularization weight in

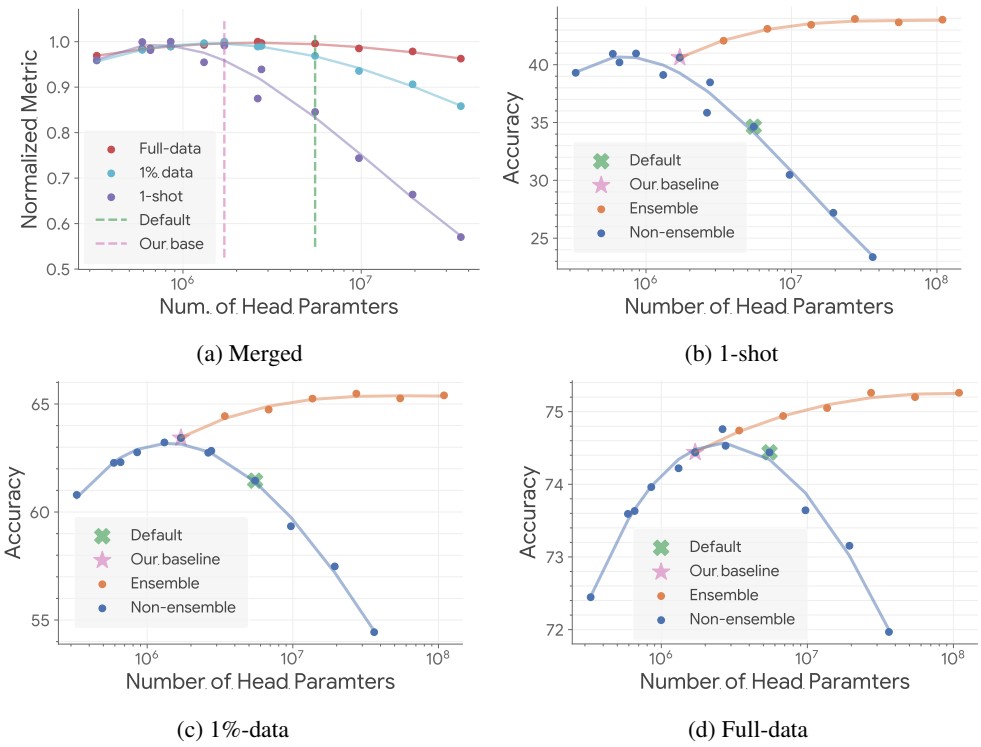

(a) Merged

(b) 1-shot

(c) 1%-data

(d) Full-data

Figure 5: **Effect of projection heads for different evaluation metrics**. We compare non-ensemble projection heads with different depths and widths as well as our $(h_\psi, \mu)$-ensembles, and evaluate linear evaluation performance with different amount of labeled data. (a) shows the comparison of normalized metrics for non-ensembles. (b)-(d) compares non-ensemble and $(h_\psi, \mu)$-ensembles by unnormalized metrics. 'Default' denotes the default projection heads used in many SSL methods. See analysis in Appx. C.2 for details.

$\{1.0, 3.0, 5.0\}$. For ME-MAX combined with Sinkhorn, we followed Assran et al. (2022) and used default regularization weight of 1.0. The results are in Table 10. We observed that:

- DINO's centering operation is not as strong as other techniques, and it favours a larger teacher temperature (e.g., 0.05). It does not work well when the codebook size (4096) is not as large as the one used in the original DINO model (65536). Switching to use Sinkhorn–Knopp normalization leads to much more improved performance, and matches the performance of original DINO (Table 5) with a much smaller codebook.

- MSN's ME-MAX regularization is very effective, and leads to significant improvements over others. We also found it is sensitive to the regularization weight and teacher temperature (c.f. Table 10). However, we observed that combining ME-MAX with Sinkhorn does not work well without tuning the regularization weight (which is recommended by Assran et al. (2022)).

## C.2 EMPIRICAL STUDY OF PROJECTION HEADS

In this subsection, we systematically study the effect of projection heads for different evaluation metrics. In particular, we used DINO* ViT-S/16 as the base model and used different projection heads with (depth, width) searched over $\{2, 3, 4\} \times \{512, 1024, 2048, 4096\}$. All models are trained with 300 epochs using exact the same set of hyper-parameters. We measured the linear evaluation performance with different amount of labeled data (i.e., full-data, 1%-data, 1-shot).

In Fig. 5a, we plot different evaluation metrics (normalized respectively by the best of each) versus the number of projection head parameters. In Figs. 5b to 5d, we plot each unnormalized evaluation metric respectively for different heads as well as our $(h_\psi, \mu)$-ensembles. Our key findings are:

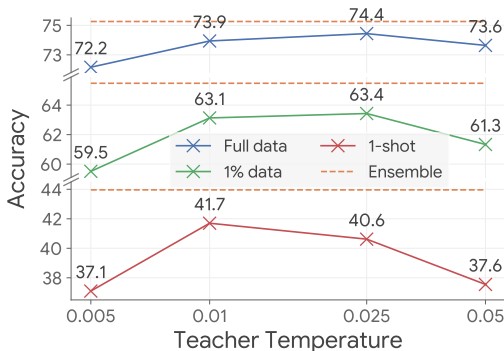

Figure 6: **Effect of teacher temperature** for non-ensemble DINO*. DINO* with a lower temperature can achieve better few-shot performance, but still under-performs our ensemble method (DINO*-ENT with 16 heads, orange lines). DINO* ViT-S/16 trained for 300 epochs is used and $\tau = 0.025$ is used for DINO*-ENT.

- The projection head has a relatively larger impact on few-shot evaluation metrics, as reflected by the relative magnitudes of different metrics in Fig. 5a. An too powerful non-ensemble projection head *significantly hurts the label efficiency* of learned representations, reflected by a much larger drop in few-shot evaluation performance (up to 18 p.p. for 1-shot, 9 p.p. for 1%-data). This result is also partially observed in Chen et al. (2020b), where they found that probing from intermediate layers of projection heads (which can be viewed as using a shallower head) could improve the semi-supervised learning (1%-/10%) results.

- The optimal projection head for different metrics can differ a lot. A weaker head improves label efficiency (few-shot performance), while a stronger (but not too strong) head improves linear decodability. As a result, the default projection head (3/2048) that is widely used in SimCLR v2 (Chen et al., 2020b), DINO (Caron et al., 2021), iBOT (Zhou et al., 2022), MSN (Assran et al., 2022), etc., does not perform well in few-shot evaluations (as shown by the green cross denoted as 'Default'), probably because it is selected by full-data evaluation metrics.

- There exist some projection heads that performs decently well on all evaluation metrics, e.g., the baseline model (3/1024) used in our experiments (pink star denoted as 'Our base').

- Compared to naively tuning projection head architectures, our $(h_\psi, \mu)$-ensembles (orange curves in Figs. 5b to 5d) consistently improve all metrics with different amount of labeled data, despite it also increases the number of parameters in projection heads. Our $(h_\psi, \mu)$-ensembles outperform all non-ensembles, which also include the counterparts of probing from intermediate layers from the a deeper head (i.e., shallower heads).

## C.3   EMPIRICAL STUDY OF $(h_\psi, \mu)$-ENSEMBLES

**Are the gains of ENT purely from sharper teacher predictions?**   Our ENT strategy assigns higher weights to the heads that predict with lower entropies, thus effectively uses sharper teacher predictions as the targets. One may be curious about how this effect accounts for the gains of the ENT strategy. We empirically answer this question by studying the non-ensemble baseline that uses a sharper teacher predictions in a *data-independent* manner (in contrast to ENT, which uses *data-dependent* entropy weights). Specifically, we compare the non-ensemble DINO* that use different teacher temperature $\tau \in \{0.005, 0.01, 0.025, 0.05\}$ and also our DINO*-ENT (16) with $\tau = 0.025$, as shown in Fig. 6. We find that the teacher temperature has a big impact on evaluation results especially for few-shot evaluation. Compared to our default baseline that uses $\tau = 0.025$, a lower temperature (e.g., $\tau = 0.01$) can indeed improve the 1-shot performance (at the cost of worse full-data performance). However, an too low temperature ($\tau = 0.005$) will hurt the performance. Our DINO*-ENT (16) consistently outperform all the baselines, which implies the importance of selecting sharper teacher predictions in a data-dependent manner.

Table 11: **Full table of Table 1** including all metrics for comparing different ensemble strategies. ENT and PROB significantly improves over the non-ensemble baseline, while UNIF leads to no gains. Ensembling the whole projection head works the best. All models are DINO* ViT-S/16 trained for 300 epochs. The means and standard deviations over 3 initialization seeds for all evaluation results are reported.

| How | Where | | Few-shot | | | | Full-data | |
|---|---|---|---|---|---|---|---|---|
| | Proj. Head | Codebook | 1 | 2 | 5 | ~13 (1%) | $k$-NN | Linear |
| Base | | | $40.6 \pm 0.2$ | $49.8 \pm 0.2$ | $57.9 \pm 0.3$ | $63.4 \pm 0.2$ | $72.3 \pm 0.1$ | $74.4 \pm 0.1$ |
| UNIF | ✓ | ✓ | $40.4 \pm 0.4$ | $49.5 \pm 0.4$ | $57.6 \pm 0.3$ | $63.3 \pm 0.3$ | $72.2 \pm 0.2$ | $74.5 \pm 0.2$ |
| PROB | ✓ | | $39.7 \pm 0.5$ | $49.0 \pm 0.5$ | $57.4 \pm 0.4$ | $63.0 \pm 0.4$ | $72.8 \pm 0.2$ | $74.8 \pm 0.1$ |
| PROB | ✓ | ✓ | $41.9 \pm 0.3$ | $51.5 \pm 0.5$ | $59.6 \pm 0.4$ | $65.1 \pm 0.3$ | $\mathbf{73.7 \pm 0.3}$ | $\mathbf{75.4 \pm 0.1}$ |
| ENT | | ✓ | $40.6 \pm 0.4$ | $49.5 \pm 0.6$ | $58.0 \pm 0.4$ | $63.5 \pm 0.4$ | $72.1 \pm 0.3$ | $74.5 \pm 0.3$ |
| ENT | ✓ | | $43.0 \pm 0.6$ | $52.2 \pm 0.8$ | $59.7 \pm 0.7$ | $64.8 \pm 0.5$ | $72.9 \pm 0.6$ | $75.1 \pm 0.4$ |
| ENT | ✓ | ✓ | $\mathbf{44.0 \pm 0.2}$ | $\mathbf{53.0 \pm 0.5}$ | $\mathbf{60.5 \pm 0.3}$ | $\mathbf{65.5 \pm 0.1}$ | $73.2 \pm 0.1$ | $\mathbf{75.3 \pm 0.1}$ |
| ENT-ST | ✓ | ✓ | $40.0 \pm 0.5$ | $39.2 \pm 0.6$ | $57.3 \pm 0.5$ | $62.7 \pm 0.5$ | $71.9 \pm 0.4$ | $74.0 \pm 0.4$ |

Table 12: **Comparison of different varaints of PROB.** The PROB strategy used in our experiments performs the best. '-' in the table denotes training divergence for PROB-MAX. The experimental setup is the same as Table 11.

| How | Where | | Few-shot | | | | Full-data | |
|---|---|---|---|---|---|---|---|---|
| | Weight by | Temp. $\gamma$ | 1 | 2 | 5 | ~13 (1%) | $k$-NN | Linear |
| Base | | | $40.6 \pm 0.2$ | $49.8 \pm 0.2$ | $57.9 \pm 0.3$ | $63.4 \pm 0.2$ | $72.3 \pm 0.1$ | $74.4 \pm 0.1$ |
| PROB | student | 1 | $\mathbf{41.9 \pm 0.3}$ | $\mathbf{51.5 \pm 0.5}$ | $\mathbf{59.6 \pm 0.4}$ | $\mathbf{65.1 \pm 0.3}$ | $\mathbf{73.7 \pm 0.3}$ | $\mathbf{75.4 \pm 0.1}$ |
| PROB-TE | teacher | 1 | $41.5 \pm 0.2$ | $50.4 \pm 0.3$ | $58.3 \pm 0.3$ | $63.7 \pm 0.1$ | $72.3 \pm 0.2$ | $74.6 \pm 0.1$ |
| PROB-MAX | student | 0 | - | - | - | - | - | - |
| PROB-MAX-TE | teacher | 0 | $41.4 \pm 0.2$ | $50.3 \pm 0.3$ | $58.1 \pm 0.3$ | $63.6 \pm 0.2$ | $72.3 \pm 0.2$ | $74.5 \pm 0.2$ |

**Comparison of different ensemble strategies and variants** We present the full table of Table 1 that includes all the metrics in Table 11. The same observation holds for all metrics.

For all previous studies, we considered a specific instantiation of PROB strategy, i.e., weight by student predicted probabilities $f_{ijy} = \log s(y|\theta_j, x)$ and $\gamma = 1$, which has a nice interpretation of model average (see Sec. 3.3). We also studied different variants of the PROB strategy (see Appx. D.1),

- PROB-TE: weight by teacher $f_{ijy} = \log t_i(y|x)$ and $\gamma = 1$;
- PROB-MAX: weight by student $f_{ijy} = \log s_j(y|x)$ and $\gamma \to 0$;
- PROB-MAX-TE: weight by teacher $f_{ijy} = \log t_i(y|x)$ and $\gamma \to 0$

Table 12 compares the downstream performance for all the variants. We find that the our PROB (used in our empirical studies) performs better than other variants. Interestingly, weighting by the teacher (PROB-TE) performs worse than PROB. We conjecture that this is because the important weights turn out to give a weighted average of teacher predictions as the surrogate target that is *shared* across all students (like PROB) but does not give effective preferential treatment across students which are directly optimized (unlike PROB-TE). Furthermore, PROB-MAX which sharpens the importance weights leads to training divergence. This is probably because the student predictions have higher variance based on which sharp weights lead to unstable training. In contrast, PROB-MAX-TE which uses the (lower-variance) teacher gives reasonable results and comparable to PROB-TE.

**Number of ensembles for MSN\*** In Fig. 7a, we study the effect of increasing the number of $(h_\psi, \mu)$-ensembles for MSN\*-ENT with ViT-S/16 trained for 800 epochs. The scaling trend is similar to DINO\*-ENT (Fig. 3a) and the gains start to diminish when the number of heads increases above 8.

**Effect of ENT temperature $\gamma$ for MSN\*** Fig. 7b studies the effect of entropy weighting temperature $\gamma$ for MSN\*-ENT. We observed that MSN\* is more robust to small temperatures, and the

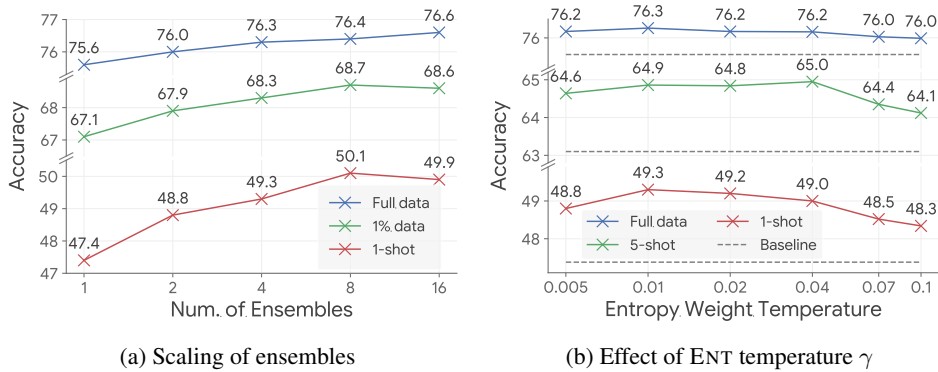

(a) Scaling of ensembles  (b) Effect of ENT temperature $\gamma$

Figure 7: **Empirical study for MSN\*-ENT.** (a) The gains by increasing the number of $(h_\psi, \mu)$-ensembles start to diminish when it is over 8 heads. (b) MSN\* prefers smaller temperature for entropy weighting than DINO\*.

best $\gamma = 0.01$ is smaller than that of DINO\* ($\gamma = 0.05$). When the temperature is too high, the performance drops as a result of under-specialization (i.e., less diversity) as with DINO\*.

## C.4 ANALYZING $(h_\psi, \mu)$-ENSEMBLE DIVERSITY

**Visualizing $(h_\psi, \mu)$-ensemble similarity**  We analyze the diversity between different heads by visualizing the similarity matrix between their codes. Directly measuring the similarity between codes in two heads could not work, because 1) they may live in different subspaces because of the ensembled projection heads; 2) they may not align in the natural order but in a permuted order.

Therefore, we seek to align codes between different heads by how they are effectively used to *'cluster'* the data. In particular, we use a set of randomly sampled inputs $\left\{x^i\right\}_{i \in [b]}$ of size $b = 51200$ to obtain an empirical code assignment matrix $A^j \in \mathbb{R}^{b \times c}$ for each $(h_\psi, \mu)$-ensemble $j \in [m]$, where the $i$-th row of $A^j$ corresponds to the teacher predictions $t_j(Y|x^i)$. For the $k$-th code in the head $j$, we extract the $k$-th column from $A^j$ (i.e., its empirical assignment) as its embedding. For two codes, we measure their similarity by the cosine similarity between their embeddings. For a pair of heads $j$ and $j'$, we align their codes using the Hungarian algorithm (Kuhn, 1955) to maximize the sum of cosine similarity. After that, we plot the similarity matrix which is aligned and reordered by the similarity value on the diagonal (in an descending order). Note that it is not necessary to do the alignment procedure for the PROB strategy since it is naturally aligned because of the direct distribution averaging over $(h_\psi, \mu)$-ensembles, but we did for fair comparison with other strategies.

We applied the same procedure for different ensemble weighting strategies using DINO\* with 4 $(h_\psi, \mu)$-ensembles. We randomly picked a pair of heads and visualize the similarity matrix before (top row) and after (bottom row) the alignment-reordering setup in Fig. 8. We found that before the alignment procedure, the similarity matrix of the PROB strategy already mostly aligns because it explicitly introduces code correspondence between different heads. Furthermore, by analyzing the similarity decay pattern on the diagonal, it is clear that ENT learns the most diverse $(h_\psi, \mu)$-ensembles while UNIF learns the least ones, which may explain the difference of their empirical performance. For completeness, we also include the visualization of aligned similarity matrices for all pairs of heads in Figs. 9 to 11, the observations are the same.

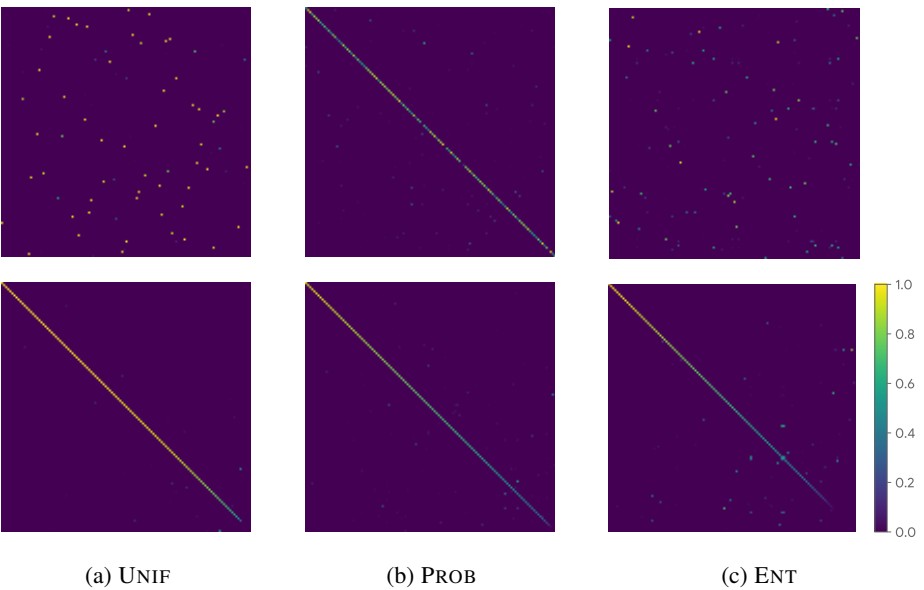

(a) UNIF          (b) PROB          (c) ENT

Figure 8: **Visualization of** $(h_\psi, \mu)$**-ensemble diversity.** ENT learns the most diverse $(h_\psi, \mu)$-ensembles while UNIF learns the least ones. We visualize the code similarity matrix between a pair of randomly selected projection heads. Top row shows the original similarity matrix (i.e., in natural order) and the bottom row shows the aligned similarity matrix which aligns codes by empirical assignment probabilities. DINO* ViT-S/16 with 4 heads is used. Best viewed in color.

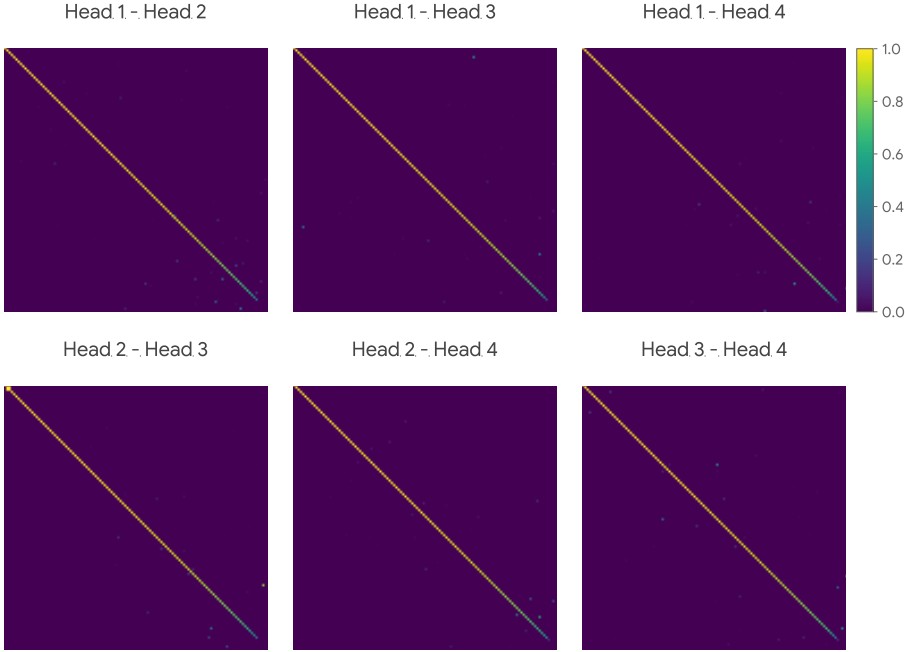

Figure 9: **Visualization of** $(h_\psi, \mu)$**-ensemble diversity between all pairs of heads for DINO*-UNIF.** The UNIF strategy does not learn diverse $(h_\psi, \mu)$-ensembles. DINO* with ViT-S/16 and 4 heads is used. Best viewed in color.

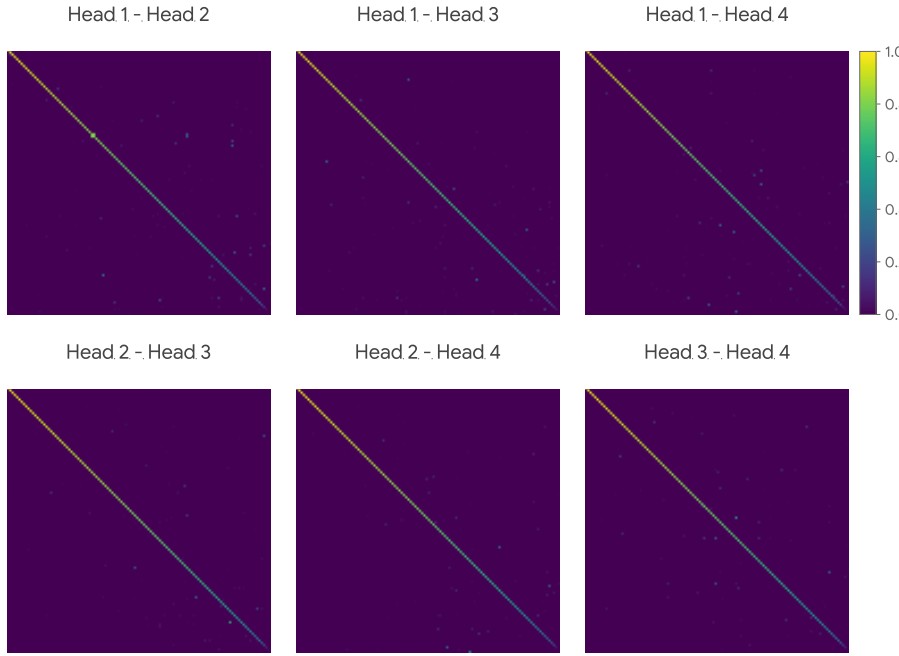

Figure 10: **Visualization of $(h_\psi, \mu)$-ensemble diversity between all pairs of heads for DINO\*-PROB.** The PROB strategy learns more diverse $(h_\psi, \mu)$-ensembles than UNIF. DINO\* with ViT-S/16 and 4 heads is used. Best viewed in color.

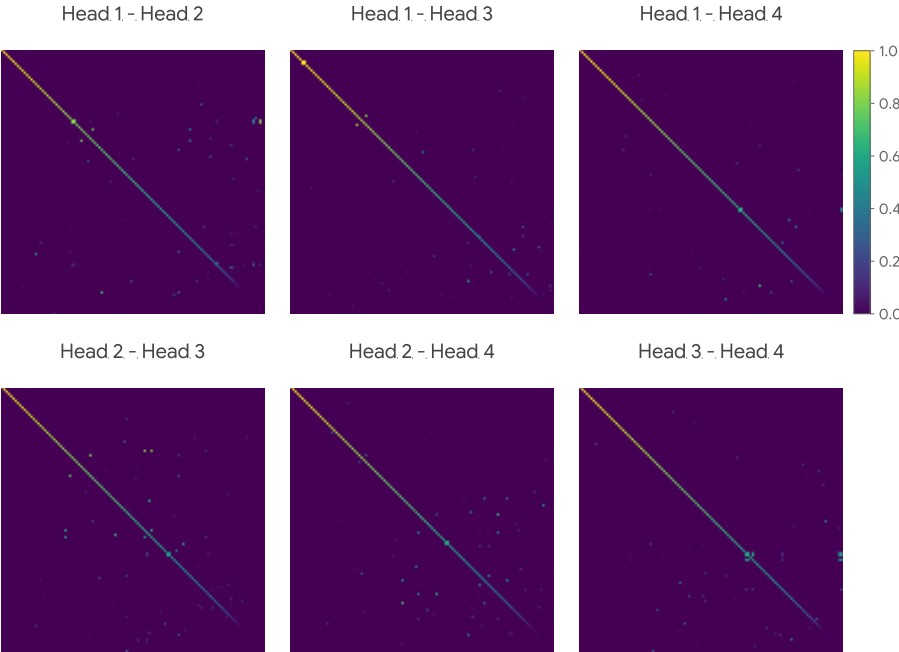

Figure 11: **Visualization of $(h_\psi, \mu)$-ensemble diversity between all pairs of heads for DINO\*-ENT.** The ENT strategy learns the most diverse $(h_\psi, \mu)$-ensembles. DINO\* with ViT-S/16 and 4 heads is used. Best viewed in color.

# D ANALYSIS

## D.1 DERIVATIONS

In this subsection, we provide derivations for some non-trivial losses that we explore within our framework.

Recall that our weighted cross-entropy loss is of the form,

$$\mathcal{L}_n(\theta) = \frac{1}{n} \sum_{x \in \mathcal{D}_n} \sum_{i,j \in [m]} \mathsf{H}^{\times}[w_{ijY} \odot t_i(Y|x), s(Y|\theta_j, x)] \tag{15}$$

$$= \frac{1}{n} \sum_{x \in \mathcal{D}_n} \sum_{i,j \in [m]} \sum_{y \in \mathcal{Y}} w_{ijy} t_i(y|x) \log s(y|\theta_j, x) \tag{16}$$

$$\text{where} \quad w_{ijy} = \mathrm{softmax}\left(\left\{\tfrac{1}{\gamma} f_{ijy}(\mathrm{stopgrad}(\theta), x) : i, j \in [m]\right\}\right). \tag{17}$$

Fuethermore, observe that,

$$\nabla_\theta \sum_{i,j \in [m]} \mathsf{H}^{\times}[w_{ijY} \odot t_i(Y|x), s(Y|\theta_j, x)] = \sum_{i,j \in [m]} \int_{\mathcal{Y}} w_{ijy} t_i(y|x) \nabla_\theta \log s(y|\theta_j, x) \mathrm{d}y. \tag{18}$$

This indicates that the proposed weighted ensemble SSL loss is simply a reweighted log-likelihood loss. We use this fact in our derivation of probability weighting (PROB) loss.

**Uniform weighting (UNIF)**  Our UNIF strategy in Eq. (6) uses $f_{ijy} = \log \delta(i - j)$ which gives $w_{ijy} = \frac{1}{m} \delta(i - j)$ (for any choice of $\gamma$), thus the loss,

$$\mathcal{L}_n^{\mathrm{UNIF}}(\theta) = \frac{1}{n} \sum_{x \in \mathcal{D}_n} \sum_{i,j \in [m]} \sum_{y \in \mathcal{Y}} \frac{1}{m} \delta(i - j) t_i(y|x) \log s(y|\theta_j, x) \tag{19}$$

$$= \frac{1}{n} \sum_{x \in \mathcal{D}_n} \frac{1}{m} \sum_{i \in [m]} \mathsf{H}^{\times}[t_i(Y|x), s(Y|\theta_i, x)] \tag{20}$$

This loss assigns equal weights to $m$ pairs of pairwised student/teacher.

An straightforward generalization is to assign equal weights to all possible pairs ($m^2$) of student/teacher with $f_{ijy} = 0$ and $w_{ijy} = \frac{1}{m^2}$, which gives the UNIF-ALL loss,

$$\mathcal{L}_n^{\mathrm{UNIF\text{-}ALL}}(\theta) = \frac{1}{n} \sum_{x \in \mathcal{D}_n} \frac{1}{m^2} \sum_{i,j \in [m]} \mathsf{H}^{\times}[t_i(Y|x), s(Y|\theta_j, x)], \tag{21}$$

**Probability weighting (PROB)**  Recall our PROB loss in Eq. (7) has the form,

$$\mathcal{L}_n^{\mathrm{PROB}}(\theta) = \frac{1}{n} \sum_{x \in \mathcal{D}_n} \mathsf{H}^{\times}\left[\frac{1}{m} \sum_{i \in [m]} t_i(Y|x), \frac{1}{m} \sum_{j \in [m]} s(Y|\theta_j, x)\right]. \tag{22}$$

We derive its equivalence with our general loss with $f_{ijy} = \log s(y|\theta_j, x)$ and $\gamma = 1$ in terms of the gradients,

$$\nabla_\theta \mathcal{L}_n^{\mathrm{PROB}}(\theta) = \frac{1}{m} \sum_{i \in [m]} \int_{\mathcal{Y}} t_i(y|x) \log \frac{1}{m} \sum_{j \in [m]} s(y|\theta_j, x) \mathrm{d}y \tag{23}$$

$$= \frac{1}{m} \sum_{i \in [m]} \int_{\mathcal{Y}} t_i(y|x) \nabla_\theta \log \frac{1}{m} \sum_{j \in [m]} s(y|\theta_j, x) \mathrm{d}y \tag{24}$$

$$= \frac{1}{m} \sum_{i \in [m]} \int_{\mathcal{Y}} t_i(y|x) \frac{\frac{1}{m} \sum_{j \in [m]} \nabla_\theta s(y|\theta_j, x)}{\frac{1}{m} \sum_{j \in [m]} s(y|\theta_j, x)} \mathrm{d}y \tag{25}$$

$$= \frac{1}{m} \sum_{i \in [m]} \int_{\mathcal{Y}} t_i(y|x) \frac{\frac{1}{m} \sum_{j \in [m]} s(y|\theta_j, x) \nabla_\theta \log s(y|\theta_j, x)}{\frac{1}{m} \sum_{j' \in [m]} s(y|\theta_{j'}, x)} \mathrm{d}y \tag{26}$$

$$= \frac{1}{m} \sum_{i,j \in [m]} \int_{\mathcal{Y}} t_i(y|x) \frac{s(y|\theta_j, x)}{\sum_{j' \in [m]} s(y|\theta_{j'}, x)} \nabla_\theta \log s(y|\theta_j, x) \mathrm{d}y \tag{27}$$

$$= \nabla_\theta \frac{1}{m} \sum_{i,j \in [m]} \mathsf{H}^\times [w_{ijY} \odot t_i(Y|x), s(Y|\theta_j, x)] \tag{28}$$

where $w_{ijy} = \frac{s(y|\theta_j, x)}{\sum_{j' \in [m]} s(y|\theta_{j'}, x)}$ (or equivalently, $f_{ijy} = \log s(y|\theta_j, x)$ and $\gamma = 1$). The last equality is because $w_{ijy}$ is stopped gradient with respect to $\theta$. This is the same analysis as done in Burda et al. (2016). The above formation establishes the equivalence of gradients between two losses, which implies the same behavior (e.g., optimum) using gradient-based optimization, as the common practice of deep learning.

We also generalize this loss to some variants which we explore in Table 12. A "dual" variant is to use teacher predictions $f_{ijy} = \log t_i(y|x)$ instead of student ones; this implies $w_{ijy} = \frac{t_i(y|x)}{\sum_{i' \in [m]} t_{i'}(y|x)}$ and the PROB-TE loss,

$$\mathcal{L}_n^{\text{PROB-TE}}(\theta) = \frac{1}{n} \sum_{x \in \mathcal{D}_n} \sum_{i,j \in [m]} \sum_{y \in \mathcal{Y}} \frac{t_i(y|x)}{\sum_{i' \in [m]} t_{i'}(y|x)} t_i(y|x) \log s(y|\theta_j, x). \tag{29}$$

Note that this simply reduces to use a weighted teacher predictions $\frac{t_i(y|x)}{\sum_{i' \in [m]} t_{i'}(y|x)} t_i(y|x)$ as the surrogate target that is shared across all students.

Another generalization is to use "hard" weighting, i.e., $\gamma \to 0$, which gives the PROB-MAX loss that only assigns weight to the most confident student,

$$\mathcal{L}_n^{\text{PROB-MAX}}(\theta) = \frac{1}{n} \sum_{x \in \mathcal{D}_n} \sum_{i,j \in [m]} \sum_{y \in \mathcal{Y}} w_{ijy} t_i(y|x) \log s(y|\theta_j, x) \tag{30}$$

$$\text{where} \quad w_{ijy} = \delta(i - i^*)\delta(j - j^*), \quad (i^*, j^*) = \arg\max_{ij} f_{ijy}, \forall y \in \mathcal{Y}. \tag{31}$$

This loss reduces to a generalization of multiple choice learning (Guzman-Rivera et al., 2012) used in multi-headed networks (Lee et al., 2015) in our ensemble SSL setup. Similarly we can also derive the dual variant of it that uses the teacher predictions, which is omitted here for brevity.

**Entropy weighting (ENT)** The derivation of ENT loss in Eq. (9) is similar to the UNIF loss but applies an entropy weights. Recall that we use $f_{ijy} = -\mathsf{H}[t_i(Y|x)] + \log \delta(i - j)$, which gives $w_{ijy} = \text{softmax}_i(\{-\frac{1}{\gamma} \mathsf{H}[t_{i'}(Y|x)] : i' \in [m]\})$ and,

$$\mathcal{L}_n^{\text{ENT}}(\theta) = \frac{1}{n} \sum_{x \in \mathcal{D}_n} \sum_{i \in [m]} \text{softmax}_i(\{-\frac{1}{\gamma} \mathsf{H}[t_{i'}(Y|x)] : i' \in [m]\}) \mathsf{H}^\times [t_i(Y|x), s(Y|\theta_i, x)]. \tag{32}$$

One can also generalizes it to its dual variant which uses the student entropies, i.e., $f_{ijy} = -\mathsf{H}[s(Y|\theta_j, x)] + \log \delta(i - j)$, which gives the ENT-ST loss,

$$\mathcal{L}_n^{\text{ENT-ST}}(\theta) = \frac{1}{n} \sum_{x \in \mathcal{D}_n} \sum_{i \in [m]} \text{softmax}_i(\{-\frac{1}{\gamma} \mathsf{H}[s(Y|\theta_{i'}, x)] : i' \in [m]\}) \mathsf{H}^\times [t_i(Y|x), s(Y|\theta_i, x)]. \tag{33}$$

## D.2 RELATING SOME LOSSES

Here, we relate some losses derived above. Specifically, we relate the uniform weighting (UNIF, UNIF-ALL) and probability weighting (PROB) in Appx. D.2.1, and relate entropy weighting (ENT) and variance weighting in Appx. D.2.2.

### D.2.1 UNIFORM & PROBABILITY WEIGHTING

We first establish the relation between UNIF and PROB using the joint convexity of unnormalized KL divergence and the fact that our weighted cross-entropy loss is a weighted unnormalized KL divergence up to some constant in $\theta$. In particular, the joint convexity of unnormalized KL divergence can be shown by combining the facts that Csiszàr $f$-divergences are jointly convex (Proposition 1 in Dragomir (2013)) and unnormalized KL divergence corresponds to the convex generator, $f(u) = u \log u - u + 1$, as required by the proposition.

First, our weighted cross-entropy loss is unnormalized KL divergence up to some constant in $\theta$:

$$\mathcal{L}_n^{\text{UNIF}}(\theta) = \frac{1}{n} \sum_{x \in \mathcal{D}_n} \frac{1}{m} \sum_{i \in [m]} \mathsf{K}[t_i(Y|x), s(Y|\theta_i, x)] + \text{constant} \tag{34}$$

$$\mathcal{L}_n^{\text{PROB}}(\theta) = \frac{1}{n} \sum_{x \in \mathcal{D}_n} \mathsf{K}\left[ \frac{1}{m} \sum_{i \in [m]} t_i(Y|x), \frac{1}{m} \sum_{j \in [m]} s(Y|\theta_j, x) \right] + \text{constant} \tag{35}$$

Therefore, the joint convexity of (unnormalized) KL divergence directly implies an ordering of the loss up to some constant in $\theta$, i.e.,

$$\mathcal{L}_n^{\text{PROB}} \leq \mathcal{L}_n^{\text{UNIF}} \tag{36}$$

Furthermore, we can also relate PROB and UNIF-ALL using the fact that the (unnormalized) cross-entropy $\mathsf{H}^\times[p(X), q(X)]$ is linear in the first argument $p$ but convex in the second argument $q$, which implies,

$$\mathcal{L}_n^{\text{PROB}} \leq \mathcal{L}_n^{\text{UNIF-ALL}} \tag{37}$$

### D.2.2 ENTROPY & VARIANCE WEIGHTING

Suppose $p(X)$ is a discrete distribution (normalized) on $\mathcal{X} = [c]$. It can be shown that,

$$\mathsf{H}[p(X)] \leq \tfrac{1}{2} \log \left( \mathsf{Var}_p[X] + \tfrac{1}{12} \right) + \tfrac{1}{2} \log (2\pi e) \tag{38}$$

where $\mathsf{Var}_p[X] = \sum_{x \in [c]} p(x)(x - \mu)^2$ and $\mu = \mathsf{E}_p[X] = \sum_{x \in [c]} p(x)x$ (Theorem 9.7.1, Cover & Thomas (1999)). Note, a tighter bound (Mow, 1998) also exists but it places stronger restrictions on $p$. This relationship suggests that choosing weights proportional to $\exp(-\mathsf{H}[t_i(Y|x)])$ (as in ENT) is potentially related to choosing weights proportional to weighting by variance $(\mathsf{Var}_{t_i(Y|x)}[Y] + \epsilon)^{-1/2}$ where $(\epsilon = \tfrac{1}{12})$.

