# OpenReview forum: "Weighted Ensemble Self-Supervised Learning"
_ICLR.cc/2023/Conference — ICLR 2023 poster_

### Official Review · Reviewer_37Hd · 2022-10-24

**Confidence:** 3
**Correctness:** 3
**Technical Novelty And Significance:** 2
**Empirical Novelty And Significance:** 2
**Recommendation:** 3

**Clarity, Quality, Novelty And Reproducibility:**

The paper describes clearly, and the proposed ensemble method is novel, but the experiments are insufficient, such as encoder ensemble and comparison with existing methods.



**Strength And Weaknesses:**

With the development of SSL, it is necessary to study SSL model ensemble. The paper is rich in theoretical basis, Explaining clustering-based self-supervised learning methods from the perspective of maximum likelihood distinctly. This paper proposed a method that can be effective solutions to ensemble multi models, because of ensembling only applying to the head and codebook, not the encoder. This paper is easy to follow, and experiments are enough to support the claims made in this paper.

Couple questions come to mind:

1.This method is effective but I think it's not enough to just ensemble the head and codebook, most researchers are probably more concerned how to ensemble the encoder part.

2.How extensible is this method? The method is verified in the double-branch structure (DINO/MSN) and can it be applied to the most popular MIM methods such as MAE.

3.How to acquire multiple models to ensemble and how to ensure the diversity of the models, it seems to be less described in the paper.

4.If I am not wrong, the authors only fuse different heads, and how these different sub-models are obtained, from the intermediate checkpoints, and how can ensure the diversity of these heads. In my opinion, the diversity of the head counts for then ensemble, and how about choosing other heads from even smaller backbones but with large diversity. How is the method compared with another simple method model soups [1] that ensembles the encoders of different checkpoints and do not increase the inference time.
Overall, although the results are impressive, I still wonder the rationale of the methods.
[1] Model soups: averaging weights of multiple fine-tuned models improves accuracy without increasing inference time

5. It is better to add average number in Table 3 for clarify.


**Summary Of The Paper:**

Model ensembling is a common technology to further boost performance. While most paper study how to ensemble on supervised models, this paper focus on self-supervised models (SSL). Then, this paper discusses where to ensemble and how to ensemble the models training from SSL, proposing a downstream-efficient ensemble method based on weighted cross-entropy objectives. The results on few-shot learning such as imagenet-1k (1%) improve baselines of SSL method (DINO/MSN).

**Summary Of The Review:**

Model ensembling for SSL model is an interesting idea to explore, however, it is still in the preliminary stage, and the above mentioned methods and experiments need to be further supplemented.

---

> ### Author Response · Authors · 2022-11-15
> **Minor**
>
> > *”It is better to add average number in Table 3 for clarify.”*
>
> Thanks for the suggestion! We have updated Table 3 with averages for clarity.

---

> ### Author Response · Authors · 2022-11-15
> **We would like more objective and grounded feedback from the reviewer**
>
> We hope the above clarifications could help the reviewer correctly understand our method and re-evaluate our paper. We note that there were some more *subjective* statements in the initial review:
>
> > *”I think it's not enough to just ensemble the head and codebook, most researchers are probably more concerned how to ensemble the encoder part”*
>
> > *”although the results are impressive, I still wonder the rationale of the methods”*
>
> We hope the reviewer could re-evaluate these comments based on our responses, and if possible, provide more clarifications for them or provide objective feedback.

---

> ### Author Response · Authors · 2022-11-15
> **Our method is applicable to other SSL methods like MAE**
>
>
> > *”How extensible is this method? The method is verified in the double-branch structure (DINO/MSN) and can it be applied to the most popular MIM methods such as MAE.”*
>
> Our ensemble method is applicable to many SSL methods like MAE beyond the two we explored. We would like to refer the reviewer to [our response to reviewer pJds](https://openreview.net/forum?id=CL-sVR9pvF&noteId=8q-T3wyqR7) and our discussion about the generalization of our method to other SSL methods included in the Conclusion (Sec. 6).

---

> ### Author Response · Authors · 2022-11-15
> **Our method is not comparable but complementary to encoder ensemble and “post-finetuning” ensemble methods**
>
> > *”How is the method compared with another simple method model soups [1] that ensembles the encoders of different checkpoints and do not increase the inference time”*
>
> > *”the experiments are insufficient, such as encoder ensemble and comparison with existing methods”*
>
> Since our method applies to the SSL training stage to directly improve representation quality while encoder ensembling or other “post-finetuning” ensemble methods like “model soup” aggregate multiple trained encoders or finetuned downstream models for evaluation, our method is not comparable but complementary to them. We would like to refer the reviewer to [our response to reviewer 8Pcm](https://openreview.net/forum?id=CL-sVR9pvF&noteId=IfUgHYD_r00) for a detailed discussion.

---

> ### Author Response · Authors · 2022-11-15
> **Clarification: Our method ensembles multiple heads to facilitate the training of a single encoder, rather than ensembling multiple SSL models post training**
>
> From the reviewer's feedback, there seems to be a crucial misunderstanding of our method which seems to lead to most of the reviewer’s concerns and questions.
>
> In particular, in the reviewer’s summary of our paper, our method is described as:
>
> > *”this paper discusses where to ensemble and how to* ***ensemble the models training from SSL****”*
>
> This is not correct. As shown in Fig. 2 and described in Sec. 3.1, our method ***jointly trains*** an ensemble of heads and/or codebooks with a single shared encoder. In contrast with traditional “post-training” ensembling that uses multiple trained models *for evaluation*, our ensembles *are NOT used for downstream evaluation* (because they are thrown away then), but are only used during training to “facilitate the learning of a single representation encoder” (as stated in the Introduction).
>
> We have further clarified this key difference in Sec 3.1 and the Introduction. We have also clarified the role of the ensemble parts in SSL in Sec. 2.
>
> &nbsp;
>
> *We believe that most of the reviewer’s concerns and questions can be naturally addressed once our method is understood correctly:*
>
> > *”How to acquire multiple models to ensemble…, it seems to be less described in the paper.”*
>
> > *”If I am not wrong, the authors only fuse different heads, and how these different sub-models are obtained, from the intermediate checkpoints… ”*
>
> As clarified above, our method does not acquire multiple models to ensemble as traditional ensembling, but trains a single encoder jointly with multiple ensembled projection heads/codebooks that are thrown away during evaluation.
>
> > *”how to ensure the diversity of the models, it seems to be less described in the paper.”*
>
> > *”how can ensure the diversity of these heads. In my opinion, the diversity of the head counts for then ensemble”*
>
> We refer the reviewer to our empirical analysis in Fig. 4 for the diversity of different heads. We did observe that the diversity of heads matters: “the choice of weighting schemes critically impacts ensemble diversity, and that greater ensemble diversity correlates with improved downstream performance” (as stated in the Abstract and Introduction).
>
> > *”how about choosing other heads from even smaller backbones but with large diversity”*
>
> Since our method does not “choose” heads from different pretrained models, this suggestion does not apply. Note that we also mentioned in Sec. 3.1 and the Conclusion that “studying heterogeneous architectures… is left for future work“.

---

> ### Author Response · Authors · 2022-11-25
> **Gentle Reminder**
>
> Dear reviewer,
>
> Thank you again for your time reviewing our paper. We would appreciate it if you could confirm that our responses address your concerns. In particular, we tried to clarify a critical misunderstanding about our method that might have significantly impacted your evaluation of our paper. We believe that we clarified the confusion and hope you could re-evaluate our paper. We would also be happy to engage in further discussions to address any other questions that you might have.
>
> Best regards,
>
> Paper3294 Authors

---

### Official Review · Reviewer_wYk7 · 2022-10-24

**Confidence:** 4
**Clarity, Quality, Novelty And Reproducibility:** Solid contribution.
**Correctness:** 3
**Technical Novelty And Significance:** 3
**Empirical Novelty And Significance:** 3
**Recommendation:** 6

**Strength And Weaknesses:**

## Strength
(1) The proposed method is straightforward and easy to implement with compelling results.

(2) I believe this paper includes some important and thorough experimental work on ensemble of self-supervised models.

(3) Detailed hyperparameters and training configurations are provided, improving the reproducibility.

## Weakness
(1) The ensembling of the projection head and codebook is a natural extension of the classifier ensemble in supervised models. It is not too surprising to see a performance improvement.

(2) This paper mainly evaluated ensemble on transfer learning, while the performance on upstream tasks (Imagenet) has been overlooked.

**Summary Of The Paper:**

This paper systematically studies the ensemble of self-supervised learning. Ensemble has been one of the oldest time-proven strategies in machine learning and has been well-studied in supervised learning. Here, the authors introduce ensemble into the field of SSL. They mainly consider ensembling the projection head and codebook and leaving the encode part as future work. This proposed method is straightforward with rich analysis and proper results.

**Summary Of The Review:**

I believe this paper, albeit short on technical novelty, does provide a sufficiently novel and important empirical evaluation and details analysis of self-supervised learning ensembling methods and could be accepted - assuming the authors can address the issues I have detailed in my review.

---

> ### Author Response · Authors · 2022-11-15
> **Clarification: Our empirical evaluations are mostly on ImageNet rather than transfer learning**
>
> > *“This paper mainly evaluated ensemble on transfer learning, while the performance on upstream tasks (Imagenet) has been overlooked.”*
>
> We respectfully point out that this is not true. Most of our experiments are on ImageNet, with the exception of those in Table 3 (transfer learning); this is stated in the paper: “We compared models trained with and without our $(h_\psi,\mu)$-ensembles by measuring various evaluation metrics on ImageNet-1K”.
>
> We have also made the evaluation dataset more clear in the captions of Table 1 & 2. If the confusion remains, we would like the reviewer to let us know where the confusion comes from, which will help us further clarify the experimental setup.

---

> ### Author Response · Authors · 2022-11-15
> **Our method is tailored for SSL and inherently different from traditional ensembling**
>
> > *“The ensembling of the projection head and codebook is a natural extension of the classifier ensemble in supervised models. It is not too surprising to see a performance improvement.”*
>
> We would like to clarify that our method is NOT a “natural extension” of classifier ensemble in supervised learning. In particular, our method ensembles the heads/codebooks that are not used for evaluation but for improving the learning of non-ensembled representation encoder *during training*, which is inherently different from traditional ensembling where the ensembles are typically obtained *post training* and used for evaluation. We have further clarified this in Sec. 3.1 and the Introduction.
>
> Although ensembling is a well-known technique for improving the evaluation performance of a single model, **we demonstrated that, for models with throw-away parts such as the projection heads in SSL, ensembling these parts can improve the learning of the non-ensembled part (i.e., the representation encoder) and also achieve significant gains in downstream evaluation without introducing extra evaluation cost**. Therefore, the performance gains could still be quite surprising. We have included this discussion in our Conclusion (Sec. 6).
>
> We also note that the **objective contributions** of our paper are the mathematical framework and the empirical performance gains of our approach, while any surprise or lack thereof is a purely **subjective** evaluation criteria that may not be universal (some readers may be surprised, others may not be) and may not be a reflection of the quality of the work.

---

> ### Author Response · Authors · 2022-11-25
> **Gentle Reminder**
>
> Dear reviewer,
>
> Thank you again for your time reviewing our paper. We would appreciate it if you could confirm that our responses address your concerns. We would also be happy to engage in further discussions to address any other questions that you might have.
>
> Best regards,
>
> Paper3294 Authors

---

### Official Review · Reviewer_8Pcm · 2022-10-24

**Confidence:** 3
**Correctness:** 3
**Technical Novelty And Significance:** 3
**Empirical Novelty And Significance:** 4
**Recommendation:** 8

**Clarity, Quality, Novelty And Reproducibility:**

- Clarity: The paper is overall clearly written.
- Quality and Reproducibility: The paper includes quite comprehensive experimental results, especially with careful treatments on the baseline methods. Experimental details are provided in the paper and the authors promise to release code later on. The results looks convincing and should be reproducible.
- Novelty: Ensembling for SSL models, especially in the pre-training stage, does not seem to have been studied in the literature. The proposed ensembling method and weighting scheme appears to be novel and more importantly, lead to decent performance improvements.

**Strength And Weaknesses:**

Strengths:
- This could be a pretty timely work that explores ensembling for self-supervised learning models. I like the idea of only ensembling the non-representation part as that incurs little training overhead, plus no extra cost for downstream tasks.
- I appreciate that the authors included the analysis on the ensemble diversity. This gives a deeper understanding of the effectiveness of different weighting schemes, and could potentially guide the designs of future weighting schemes.
- The paper includes thorough analysis and ablations on the proposed methods, and shows strong empirical results compared to the baselines and SOTA.

Weaknesses:
- The connection between the Maximum Likelihood view of SSL and the proposed ensemble method design is not very clear to me. The presentation could be improved with stronger connections between Sec.2 and Sec.3.
- While the paper does a good job in evaluating a variety of ablations on the proposed ensemble method, it seems to lack other possible baselines for ensembling, e.g., instead of only ensembling the projection head, one baseline is to consider ensembling the base models. It’d be good to compare this baseline to understand whether there is a trade-off between the benefits of ensembling and the cost for extra parameters. In general, the paper should include comparisons to the other related ensemble baseline mentioned in Sec. 4.

Other questions/comments:
- Can one use the proposed ensembling method on pre-trained SSL models (e.g. DINO)? What’s the performance difference of that to applying ensembling in the pre-training from scratch? I think this is a relevant question to look into since there are already a lot of publicly available pre-trained SSL models, and it’d be best if the ensembling technique can be applied on top of these pre-trained models and still offer improvements.
- How does one set $\gamma$ for ENT in practice? From Figure 3.b, it seems to affect the performance pretty much.


**Summary Of The Paper:**

The paper studies the benefits of ensembling for self-supervised learning methods. The paper proposes to ensemble the “non-representation” part of the SSL models, and proposes different weighting schemes. Extensive experiments are conducted to analyze and evaluate the proposed ensemble method and weighting schemes, which lead to improvements over the non-ensemble models.

**Summary Of The Review:**

The paper studies a timely topic on ensembling for SSL models. The proposed method is simple and could fit with a variety of SSL models. While the paper can be made better by comparing to more related ensembling baselines to guide practical use, the current presented empirical results is quite comprehensive on its own, and the usefulness of the proposed method is well-justified by the empirical results.

---

> ### Author Response · Authors · 2022-11-15
> **Minor questions**
>
> > *“Can one use the proposed ensembling method on pre-trained SSL models (e.g. DINO)? What’s the performance difference of that to applying ensembling in the pre-training from scratch?”*
>
> Thanks for the suggestion! This is an interesting idea for extending our method. Since it is not directly relevant to the question we study in the paper, we will leave it for future work. We note that our mathematical framework (in Sec. 3) supports such an ensembling scheme, but depending on the exact setting you have in mind, our efficient ensembling approach may or may not be relevant.
>
>
> > *”How does one set γ for ENT in practice? ”*
>
> We select $\gamma$ with downstream validation performance. Although different evaluation metrics could be sensitive to $\gamma$ to varying degrees, the overall trends are similar (as shown in Fig. 3(a)); we chose to use linear evaluation with 1% data for selection.

---

> ### Author Response · Authors · 2022-11-15
> **Our method is not comparable but complementary to encoder ensembling and other “post-training/finetuning” ensemble methods**
>
> > *”one baseline is to consider ensembling the base models”*
>
> > *”In general, the paper should include comparisons to the other related ensemble baseline mentioned in Sec. 4”*
>
> We would like to emphasize the key difference of our method from encoder ensembling and other ensemble methods mentioned in Sec. 4: in our method the ensembled heads/codebooks are only used during training for improving the learning of non-ensembled representation encoder, thus incurring no downstream evaluation cost; while encoder ensembling and other ensemble methods mentioned in Sec. 4 are “post-training/finetuning” similar to traditional supervised ensembling, where multiple *trained* representation encoders or fine-tuned downstream models are aggregated for evaluation.
>
> Therefore, **our method is not comparable but complementary to them**. For example, one can ensemble multiple encoders for downstream evaluation, each of which is trained with our ensemble heads; one can also apply “model soup” ensembling to finetune the encoder trained with our method on downstream tasks.
>
> We have made this distinction more clear in Sec. 3.1 and Sec. 4.

---

> ### Author Response · Authors · 2022-11-15
> **We have improved the presentations to better connect Sec. 2 and Sec. 3**
>
> > *“The connection between the Maximum Likelihood view of SSL and the proposed ensemble method design is not very clear to me.”*
>
> We have improved the presentations of Sec. 2 and Sec. 3. In particular, the MLE formulation of the SSL losses in Sec. 2 is used as the notational basis for deriving our weighted ensemble loss in Sec. 3; our loss is a weighted cross-entropy that respects the MLE intuition. We have clarified these points in Sec. 2 & 3.

---

> ### Comment · Reviewer_8Pcm · 2022-11-22
> **Thank you for the response**
>
> Thank you to the authors for providing the responses. I agree that the proposed method is complementary to encoder ensembling. It'd be great to include, even just preliminary experiments in the final revision, to show how the proposed method can be combined with encoder ensembling as pointed out in the response below. I would remain my score as is.

---

### Official Review · Reviewer_pJds · 2022-10-31

**Confidence:** 3
**Correctness:** 3
**Technical Novelty And Significance:** 2
**Empirical Novelty And Significance:** 3
**Recommendation:** 6

**Clarity, Quality, Novelty And Reproducibility:**

- Clarity: This paper is well-organized and easy to read.
- Novelty: The proposed ensemble framework seems to be a little bit simple. But the experiments are sufficient.

**Strength And Weaknesses:**

Strength:
- This paper is well-written and easy to follow.
- The ensemble methods only incur a small training cost.
- It is interesting to see that sota SSL algorithms can be further improved by ensembling in various settings (linear eval, KNN, few-shot, etc.).

Weaknesses:
- The proposed framework only suits contrastive SSL methods (i.e., align positive pairs). Other SSL methods such as MAE can not be applied.
- When encoders are not considered for the ensemble, the possibilities for where to ensemble are not too many. The proposed ensemble framework somewhat lacks novelty. The main contribution is to test different weighting schemes.

**Summary Of The Paper:**

This paper studies how ensemble methods can improve self-supervised learning algorithms. The considerations are limited to ensembles of projection heads and codebooks but not encoders, to ensure no extra computational cost involved. The authors propose a kind of data-dependent weighted cross-entropy losses. Two sota SSL methods, DINO and MSN are considered as baselines. In various experimental settings, the ensemble methods are observed to largely improve the baselines.

**Summary Of The Review:**

Overall,  the problem this paper studies is interesting. To simplify the design and retain the computational cost, this paper is limited to ensembles of projection heads and codebooks but not encoders, which limits the technical novelty. It is interesting to see the significant improvement in performance under various settings and evaluation metrics. Therefore, I think this is a borderline paper.

---

> ### Author Response · Authors · 2022-11-15
> **Our method is novel because it is inherently different from traditional ensembling**
>
> > *”When encoders are not considered for the ensemble, the possibilities for where to ensemble are not too many. The proposed ensemble framework somewhat lacks novelty. ”*
>
> We would like to emphasize that choosing not to ensemble the encoder is a deliberate decision tailored for SSL. In particular, we demonstrated that, for models with throw-away parts--such as the projection heads in SSL--ensembling these parts can **improve the learning of the non-ensembled part** (i.e., the representation encoder) and also achieve significant gains in downstream evaluation **without introducing extra evaluation cost**; this makes our method novel and inherently different from traditional ensembling in supervised learning or encoder ensembling where the ensembles are typically obtained **post training** and used for evaluation. See also [our response to Reviewer wYk7](https://openreview.net/forum?id=CL-sVR9pvF&noteId=HIm9VKO8gD).
>
> > *”The proposed ensemble framework seems to be a little bit simple”*
>
> We consider the simplicity of our method to be a strength rather than a weakness. Given that it is easy to implement, applicable to various SSL methods, and led to SOTA gains, we are hopeful it is a useful contribution to the community.

---

> ### Author Response · Authors · 2022-11-15
> **Our method is applicable to other SSL methods like MAE**
>
> > *”The proposed framework only suits contrastive SSL methods (i.e., align positive pairs). Other SSL methods such as MAE can not be applied.”*
>
> We would like to clarify that our ensemble method is applicable to many SSL methods beyond the two we explored. Within our mathematical framework one may consider studying MAEs by ensembling multiple decoders. Since the MAE decoders are discarded during downstream evaluation, no evaluation cost is incurred as well. Nevertheless, ensembling MAE decoders are more computationally expensive than ensembling heads/codebooks, and MAE obtains inferior performance on our evaluation benchmarks, we intentionally left this for future work.
>
> We focused on clustering SSL methods like DINO/MSN because they are the state-of-the-art methods on the evaluation benchmarks; thus they serve as a good starting point for demonstrating the effectiveness of ensembling for SSL. We hope that the significant gains of our method upon this class of SSL methods and insights from our analysis will motivate more future work for extending our method to different SSL methods.
>
> We have included a discussion about the generalization of our method to other SSL methods in the Conclusion (Sec. 6).

---

> ### Author Response · Authors · 2022-11-25
> **Gentle Reminder**
>
> Dear reviewer,
>
> Thank you again for your time reviewing our paper. We would appreciate it if you could confirm that our responses address your concerns. We would also be happy to engage in further discussions to address any other questions that you might have.
>
> Best regards,
>
> Paper3294 Authors

---

> ### Comment · Reviewer_pJds · 2022-12-08
> **Thanks for the response**
>
> Thanks for the response! Some of my concerns have been addressed. However, the methods which can be ensembled are somewhat limited. Meanwhile, SOTA gains may not directly lead to a high score. Therefore, I would like to keep my score.

---

### Author Response · Authors · 2022-11-15
**Response Summary**

We thank all reviewers for their helpful feedback.

We are glad that the reviewers found our paper well-written [pJds, 8Pcm, 37Hd], the problem we studied interesting [pJds] and necessary to study [37Hd], our proposed method novel [8Pcm, 37Hd], and our empirical study sufficient [pJds, 37Hd], thorough [8Pcm, wYk7], and convincing [8Pcm]. We are also thankful that **all** the reviewers recognized the impressive empirical performance of our method with significant improvements over SOTA baselines, and reviewers [8Pcm, wYk7] recognized the reproducibility of our empirical results and our careful treatments of baselines [wYk7].

There are concerns mainly about some *additional* research questions for ensemble SSL, e.g., studying encoder ensembling [8Pcm, 37Hd] and extending our method to other SSL methods like MAE [pJds, 37Hd]. We agree that these are interesting questions that are worth studying for future work, but we do not believe their study is necessary to support the central contributions of this paper; all reviewers agree that our empirical studies sufficiently support our claims and that we demonstrated significant gains over state-of-the-art SSL methods. Nevertheless, we incorporated some additional discussions as suggested by the reviewers for studying these questions in the paper for the sake of interest for future research.

Our main changes to the paper are summarized below:
* **Highlighting the technical novelty** [pJds, wYk7]. We emphasized in the Introduction (Sec. 1) and the Conclusion (Sec. 6), that one of our novel technical contributions is to demonstrate the effectiveness of an ensemble method that is specifically tailored for SSL and inherently different from traditional ensembling. See [our response to Reviewer wYk7](https://openreview.net/forum?id=CL-sVR9pvF&noteId=HIm9VKO8gD).
* **Clarifying the key distinction between our method and encoder ensembling** [8Pcm, 37Hd]. In Sec. 3.1 and Sec. 4, we further clarified the key difference of our method from encoder ensembling and other “post-training/finetuning” ensemble methods. Crucially, our method is not comparable but complementary to these methods. See [our response to reviewer 8Pcm](https://openreview.net/forum?id=CL-sVR9pvF&noteId=IfUgHYD_r00).
* **Discussing the applicability to other SSL methods** [pJds, 37Hd]. We included a discussion of the applicability of our ensemble method to other SSL methods like MAE in the Conclusion (Sec. 6). See [our response to reviewer pJds](https://openreview.net/forum?id=CL-sVR9pvF&noteId=8q-T3wyqR7).
* **Improving presentation to better connect Sec. 2 and Sec. 3** [8Pcm]. We improved the presentation of Sec. 2 and Sec. 3 to make their connection more clear.

Our changes to the paper are highlighted in blue text. Please see individual responses below for other specific changes and more details.

---

### Decision · Program_Chairs · 2023-01-20

**Decision:**

Accept: poster

**Justification For Why Not Higher Score:**

The novelty of the paper is just marginally above the bar.

**Justification For Why Not Lower Score:**

The performance of the proposal should be considered extremely strong. This implies the significance of the paper is clearly above the bar.

**Metareview: Summary, Strengths And Weaknesses:**

The paper studied ensemble learning for self-supervised learning with the main focus on answering "where" and "how" to ensemble. The proposal is not the traditional post-training ensemble but a novel during-training ensemble. Although this type has been proposed for supervised learning, the current paper should be the first such algorithm for self-supervised learning. Since the two baselines are very strong, the performance of the proposal should be considered extremely strong. This implies the significance of the paper is clearly above the bar and thus we should accept it for publication.

**Note From Pc:**

if the above contains the word "oral" or "spotlight" please see: "oral" presentation means -> notable-top-5% and "spotlight" means -> notable-top-25%. As stated in our emails, we are disassociating presentation type from AC recommendations

**Summary Of Ac-Reviewer Meeting:**

Two reviewers, 8Pcm and wYk7, attended the meeting. We exchanged the opinions fruitfully. The novelty is marginally above the bar and the significance is clearly above the bar. We all vote for acceptance. On the other hand, the only negative reviewer, 37Hd, is not responsive at all. Therefore, I didn't take this negative review into account.